# Tversky Neural Networks: Psychologically Plausible Deep Learning with Differentiable Tversky Similarity

**Moussa Koulako Bala Doumbouya**    **Dan Jurafsky**    **Christopher D. Manning**

Department of Computer Science, 353 Jane Stanford Way; Stanford, CA 94305
{moussa, jurafsky, manning}@stanford.edu

Figure 1: NABirds and MNIST stimuli sorted from low to high salience (Equation 2) in TverskyResNet-50. More salient stimuli exhibit more object features and greater *informational content*, *familiarity*, and *goodness of form*, in alignment with human judgment (Tversky, 1977).

## Abstract

Work in psychology has highlighted that the geometric model of similarity standard in deep learning is not psychologically plausible because its metric properties such as symmetry do not align with human perception of similarity. In contrast, Tversky (1977) proposed an axiomatic theory of similarity with psychological plausibility based on a representation of objects as sets of features, and their similarity as a function of their common and distinctive features. This model of similarity has not been incorporated as a general-purpose building block in deep learning, in part because of the challenge of incorporating discrete set operations. In this paper, we develop a differentiable parameterization of Tversky's similarity that is learnable through gradient descent, and derive basic neural network building blocks such as the *Tversky projection layer*, which unlike the linear projection layer can model non-linear functions such as XOR. Through experiments with image recognition and language modeling neural networks, we show that the Tversky projection layer is a beneficial replacement for the linear projection layer. For instance, on the NABirds image classification task, a ResNet-50 with a Tversky projection layer trained from scratch achieves a 2.36 percentage point accuracy improvement over the linear layer baseline. With Tversky projection layers, GPT-2's perplexity on PTB decreases by 7.8%, and its parameter count by 34.8%. Finally, we propose a unified interpretation of both projection layer types as computing similarities of inputs to learned prototypes, along with a novel visualization technique. Crucially, Tversky's set-based representation enables the algebraic specification of *semantic fields*, which we illustrate with lexical and visual stimuli. Our work offers a new paradigm for neural networks that are not only more accurate and efficient, but also interpretable under an established theory of psychological similarity.[1]

---

[1] https://github.com/mdoumbouya/tversky-networks-iclr2026

# 1 INTRODUCTION

The geometric model of similarity is ubiquitously used in modern neural networks. Their architectures include operations that assess the similarity between objects using metric similarity functions such as the vector dot product or cosine similarity. For instance, linear projection layers, also known as *dense*, or *fully connected*, output the dot-product similarity of the input vector with the columns of its weight matrix, which are concept prototypes. Classification modules typically normalize these similarities with SOFTMAX to form valid probability distributions (Bridle, 1990; LeCun et al., 1998). Convolution layers of image recognition neural networks compute the dot product of image regions and convolution kernels, which represent prototypical patterns. LSTM gates compute dot product similarities of combinations of inputs, hidden states, and cell states with prototypical *input*, *forget*, and *output* gating patterns (Hochreiter and Schmidhuber, 1997; Graves et al., 2005; Graves, 2013). Semantic word embedding models such as word2vec (Mikolov et al., 2013a) and GloVe (Pennington et al., 2014) compute the dot-product and cosine similarity of word representations with prototypical embeddings to predict words. Neural language models apply the same classification mechanism to output discrete words or subwords at each time step (Bengio et al., 2003; Sutskever et al., 2014; Devlin et al., 2019; Radford et al., 2018; OpenAI et al., 2024). Attention weights (Bahdanau et al., 2015; Vaswani et al., 2017) are the normalized dot-product similarity of *queries* and *keys*, which are themselves obtained using linear projection layers. Geometric similarity representations have also been widely used in machine learning objective function design for over 200 years, from least squares methods (Legendre, 1806) to modern approaches such as L2 loss (Hadsell et al., 2006), Large Margin Cosine Loss (Wang et al., 2018), and Noise Contrastive Estimation (van den Oord et al., 2018; Chen et al., 2020), spanning both supervised and self-supervised learning paradigms.

However, Tversky (1977) famously challenged this geometric representation of similarity, showing its psychological implausibility due to its fundamental inability to model phenomena such as humans' asymmetric judgment of similarity. For instance, "We say that *'the son resembles the father'* rather than *'the father resembles the son'*." He addressed this problem with his axiomatic theory based on a feature matching process in which objects are represented as sets of features and their similarity is measured as a linear combination of measures of their common and distinctive features. Tversky's similarity has been shown to model human judgment more accurately than metric similarity measures in various empirical studies (Gati and Tversky, 1984; Ritov et al., 1990; Siegel et al., 1982; Rorissa, 2007). Moreover, over the past decade, several works have highlighted the limitations of the geometric model of similarity and related parallelogram model of semantic relations (Garg et al., 2015; Chen et al., 2017; Peterson et al., 2020; Zhou et al., 2022), and advocated for more psychologically principled similarity measures such as Tversky's model, which has strong empirical support.

Nevertheless, a differentiable expression of Tversky's similarity suitable as a general-purpose neural network building block has not been proposed to date, preventing its ubiquitous integration in gradient-based machine learning (Rumelhart et al., 1986; LeCun et al., 1998; Hochreiter and Schmidhuber, 1997; Vaswani et al., 2017; He et al., 2016; Radford et al., 2018; OpenAI et al., 2024). The formulation of a differentiable Tversky similarity function is non-trivial because it employs measures of set intersections and differences, which are not differentiable with respect to object features comprising those sets. To address this gap, we propose a novel representation of features as vectors of the same dimensionality as object vectors, and a dual representation of objects both as vectors and as sets, such that an object is the set of features with which it has a positive dot product. This representation of objects and features enables us to define differentiable measures of set intersections and differences, which we use to construct a differentiable Tversky similarity function suitable for deep learning. We also propose the *Tversky Projection* neural layer, analogous to the linear projection layer but computing Tversky rather than dot-product similarities of their input to prototypes.

We introduce our proposed differentiable similarity function, and basic neural network modules, along with a novel method for interpreting and visualizing deep projection layers in Section 3. Section 4 demonstrates the effectiveness of Tversky Projection layers in state-of-the-art architectures, including ResNet-50 and GPT-2, on image recognition and language modeling tasks. Our experiments show that the use of Tversky projection layers can lead to significant improvements in both accuracy and parameter efficiency. In particular, we show that the use of a Tversky Projection layer instead of the final linear projection layer in ResNet-50 can result in up to 2.36 percentage points accuracy improvement in image recognition tasks. We also show that replacing linear projection layers in the language modeling head, and attention blocks in GPT-2 can result in a 7.8% reduction in perplexity

and 34.8% reduction in parameter count. Section 4.4 provides qualitative analyses of Tversky projection layers, including results highlighting their principled explainability under Tversky's theory of similarity, which enables the algebraic specification of human-interpretable *semantic fields*. Overall the main contribution of this work is to provide a foundation for efficient and interpretable neural networks based on an established theory of psychological similarity.

## 2  RELATED WORK

Here we briefly discuss prior work related to Tversky neural networks.

**Tversky Loss for Semantic Segmentation:** Salehi et al. (2017) propose a Tversky-based loss for 3D image segmentation, treating voxel predictions and ground truths as features. Unlike their approach, our definition of features is not restricted to pixels or voxels, and is applicable in any data modality and any neural network layer.

**Semantic Similarity Networks:** Rahnama and Hüllermeier (2020) learn a semantic similarity network via a two-level architecture of binary feature extractors followed by a Tversky similarity layer, but their approach is limited to similarity learning and requires stimulus pairs annotated with similarity scores and fine-grained feature annotations. Our method uses a learnable feature bank to measure feature presence and salience, requires no such annotations, and serves as a general-purpose building block usable in any neural network.

**Mixture-of-Expert Networks:** Tversky layer indicator functions act as hard routers, making Tversky layers sparsely activated modules analogous to Mixture-of-Experts (MoE) gating networks (Jacobs et al., 1991; Shazeer et al., 2017; Fedus et al., 2022; Hwang et al., 2026). We discuss the related "dead feature" issue in Section 5.

**Prototype Visualization:** ProtoPNet (Chen et al., 2019) and ProtoTree (Nauta et al., 2021) learn latent patch representations compared against convolutional feature maps to infer classification evidence, but are restricted to convolutional patches and cannot visualize fully connected layer parameters. Our method visualizes the parameters of any projection layer, fully connected or Tversky, directly in input space at any network depth.

**Algebraic Semantic Fields:** Vector arithmetic over word representations can probe semantic concepts such as analogies (Mikolov et al., 2013b; Molino et al., 2019), but has limited expressivity, notably failing to characterize antonyms (Ali et al., 2024; Faruqui et al., 2015; Mrkšić et al., 2016; Ono et al., 2015). In contrast, our set-based representations enable the specification of arbitrary semantic fields using set-algebraic expressions in any domain and modality (Section 4.4).

## 3  METHODS

### 3.1  TVERSKY'S SIMILARITY

Tversky's model of similarity (Tversky, 1977) has emerged as an influential theoretical representation of human similarity judgment, supported by extensive empirical evidence from cognitive psychology (Tversky and Gati, 1982; Gati and Tversky, 1984; Goldstone, 1994; Medin et al., 1993; Rorissa, 2007). His work challenged the geometric model of similarity by demonstrating that humans systematically violate metric axioms such as minimality, symmetry, and triangle inequality when assessing similarity, and proposed a theoretical framework in which objects are represented as sets of features and their similarity assessment as a feature matching process. Formally, Tversky's asymmetric model of similarity of object $a$ to object $b$ defined as feature sets $A$ and $B$ is a function $F$ of their common and distinctive features. Tversky showed in his axiomatic framework that $F$ is a linear combination of measures of its set parameters, namely: the common features of $a$ and $b$, the distinctive features of $a$, and the distinctive features of $b$ (Equation 1). In Section 3.2, we introduce a differentiable parameterization of this function, making it suitable for gradient-based machine learning.

$$S(a,b) = F(A \cap B, A - B, B - A) = \theta f(A \cap B) - \alpha f(A - B) - \beta f(B - A) \tag{1}$$

## 3.2 Differentiable Tversky Similarity

This section presents our proposed differentiable parameterization of Tversky's similarity function, which is constructed with a representation of features as vectors of the same dimensionality as objects, and a dual representation of objects as vectors and as sets.

**Dual Representation of Objects as Vectors and as Sets:** Given the learnable finite universe $\Omega$ of feature vectors $f_k \in \mathbb{R}^d$, and an object represented as the vector $x \in \mathbb{R}^d$, we propose $x \cdot f_k$ to be the scalar measure of feature $f_k$ in $x$, and a second representation of $x$ as the set $X = \{f_k \in \Omega | x \cdot f_k > 0\}$ of features with which $x$ has a positive dot product.

**Salience:** Tversky hypothesized that the relative salience of stimuli, or prominence of their features, determines the direction of asymmetry in human's judgment of similarity. The less salient stimulus (e.g. the son) is assessed to be more similar to the more salient stimulus (e.g. the father) than vice versa. Following Tversky's theory and our proposed representation, the salience of features in an object $A$, which is the sum of the measures of all features present in the object is $f(A)$ (Equation 2).

$$f(A) = \sum_{k=1}^{|\Omega|} a \cdot f_k \cdot \mathbb{1}[a \cdot f_k > 0] \tag{2}$$

**Feature Set Intersections:** To measure the common features of objects $A$ and $B$, $f(A \cap B)$ (Equation 3), we propose a function $\Psi$ to aggregate measures of the features present in both $a$ and $b$, and experiment with values *min*, *max*, *product*, *mean*, *gmean* and *softmin*. This function corresponds to the *intersection reduction* hyperparameter of Tversky neural modules.

$$f(A \cap B) = \sum_{k=1}^{|\Omega|} \Psi(a \cdot f_k, b \cdot f_k) \times \mathbb{1}[a \cdot f_k > 0 \wedge b \cdot f_k > 0] \tag{3}$$

**Feature Set Difference:** $f(A - B)$ is a measure of features present in $A$ but not present in $B$ (Equation 4). We propose an alternate form of this measure that accounts for features that are present in both $A$ and $B$, but in greater amount in $A$ (Equation 5). These two measures of set difference respectively correspond to the values *ignorematch* and *substractmatch* of the *difference reduction* hyperparameter of Tversky neural modules.

$$f^i(A - B) = \sum_{k=1}^{|\Omega|} (a \cdot f_k) \times \mathbb{1}[a \cdot f_k > 0 \wedge b \cdot f_k \leq 0] \tag{4}$$

$$f^s(A - B) = f^i(A - B) + \sum_{k=1}^{|\Omega|} (a \cdot f_k - b \cdot f_k) \times \mathbb{1}[b \cdot f_k > 0 \wedge a \cdot f_k > b \cdot f_k] \tag{5}$$

## 3.3 Tversky Neural Network Modules

We propose two basic building blocks for Tversky neural networks, the *Tversky Similarity Layer*, which is analogous to metric similarity functions such as dot product or cosine similarity, and the *Tversky Projection Layer*, analogous to the contemporary *fully connected layer*.

**Tversky Similarity Layer:** This layer, formalized in Equation 6, calculates the similarity of object $a \in \mathbb{R}^d$ to object $b \in \mathbb{R}^d$. Its learnable parameters are a feature bank $\Omega$, and the $\alpha$, $\beta$ and $\theta$ scalar parameters of Tverky's contrast model of similarity (Equation 1).

$$\mathcal{S}^{\Omega,\alpha,\beta,\theta}(a,b) \colon \begin{cases} \mathbb{R}^d \times \mathbb{R}^d \longrightarrow \mathbb{R} \\ (a,b) \longmapsto \begin{bmatrix} \theta \\ -\alpha \\ -\beta \end{bmatrix} \cdot \begin{bmatrix} f(A \cap B) \\ f(A - B) \\ f(B - A) \end{bmatrix} \end{cases} \tag{6}$$

**Tversky Projection Layer:** This non-linear projection (Equation 7), calculates the similarity of its input $a \in \mathbb{R}^d$ to each prototype in the ordered set of $p$ prototypes $\Pi_i \in \mathbb{R}^d$, yielding a vector in $\mathbb{R}^p$.

$$\mathcal{P}^{\Omega,\alpha,\beta,\theta,\Pi}(a) \colon \begin{cases} \mathbb{R}^d \longrightarrow \mathbb{R}^p \\ a \longmapsto \begin{bmatrix} \mathcal{S}^{\Omega,\alpha,\beta,\theta}(a, \Pi_0) \\ \mathcal{S}^{\Omega,\alpha,\beta,\theta}(a, \Pi_1) \\ \vdots \\ \mathcal{S}^{\Omega,\alpha,\beta,\theta}(a, \Pi_{p-1}) \end{bmatrix} \end{cases} \tag{7}$$

### 3.4 INTERPRETATION OF PROJECTION LAYERS

Both the linear and Tversky projection layers output vectors in which each dimension is the similarity of the input to a prototype; however, they differ in the employed similarity function. Linear projection layers compute dot-product similarities. This is an application of the geometric model of similarity, in which the features of objects and prototypes are scalar values arranged in a Cartesian coordinate space and similarity is measured as the oriented length of the object vector's projection onto the prototype vector. Dot product similarity satisfies metric axioms such as symmetry and triangle inequality that have been empirically shown to not align with human perception, and is provably incapable of modeling asymmetric relations. Tversky projection layers compute Tversky similarity with learnable parameters prototype vectors $\Pi$, feature vectors $\Omega$, and Tversky's contrast model's weights $\alpha$, $\beta$, and $\theta$. The number of features $|\Omega|$ can be varied without affecting the output dimensionality $p = |\Pi|$.

### 3.5 TVERSKY FEATURE SHARING

Tversky feature banks and prototype banks can be shared across various layers in a neural network in semantically justifiable ways. For instance, in a Tversky GPT-2, the output projection layers in attention blocks, and the final language modeling head can all share the same feature bank $\Omega$ as this parameter is semantically compatible across all those layers: it represents token features. Tversky projection layers that are language modeling heads can also use token embeddings as token prototypes. This parameter sharing strategy is similar to *weight tying* widely used in language models employing a linear projection layer to classify output tokens (Inan et al., 2017). Multiple Tversky similarity or projection layers can share their feature or prototype bank if semantically compatible. They can also share the same feature bank while maintaining separate prototypes, or keep all their parameters separate. Our results in Section 4.2 show that Tversky feature sharing can result in a dramatic reduction of neural network parameter count while improving their performance.

### 3.6 DATA-DOMAIN VISUALIZATION OF PROJECTION LAYERS

We introduce a novel method of visualizing projection layer parameters such as the weights of fully connected layers, and the prototypes and features of Tversky projection layers to enable their interpretation in the same domain as data stimuli. Our method is based on the hypothesis that those prototypes and features are concepts that should be recognizable in the data domain, and is applicable regardless of the depth at which those projection layers are employed in deep neural networks. Our proposed visualization method consists of specifying the projection parameters as tensors of the same shape as the input data. These input-domain-specified parameters are forwarded through the neural network just like data stimuli up to the layer prior to the layer in which they are used. Their obtained vector representation is subsequently used to perform the projection. This approach significantly differs from prior approaches of interpreting neural network parameters and decisions based on the visualization of activations, optimization of input stimuli, or construction of adversarial examples (Zeiler and Fergus, 2014; Mordvintsev et al., 2015; Zhang and Zhu, 2018; Selvaraju et al., 2020; Zhang and Liang, 2020; Fan et al., 2021). While our proposed method offers the clear advantage of visualizing projection parameters, it comes with the limitation that parameters specified in data-space are typically larger in size than their original counterparts, which increases the effective number of trainable parameters. These parameters also need to be forwarded along with every data batch, which induces additional training-time computation cost (See Figure 5 in Appendix A). We use this technique to qualitatively compare the parameters of a linear projection layer and a Tversky projection layer of neural networks trained to recognize handwritten digits in our qualitative analysis experiment presented in Section 4.4, which revealed that Tversky Projection layers' parameters are far more interpretable than the ones of the contemporary fully connected layer (see Figure 3).

## 4 EXPERIMENTS

Here we demonstrate the utility of our proposed differentiable Tversky similarity function in machine learning. First, we show by construction that a single Tversky projection layer can model the XOR function and report empirical results on the same learning task (Section 4.1). Then, in Sections 4.2 and 4.3, we conduct empirical experiments with state-of-the-art neural networks for language modeling and image recognition in which we compare baseline neural networks with their counterparts

employing Tversky projection layers. Finally, in Section 4.4, we conduct qualitative analyses demonstrating the interpretability of Tversky neural networks.

### 4.1 MODELING XOR WITH A SINGLE TVERSKY PROJECTION LAYER

As a first experiment, we construct a single Tversky projection layer that computes the XOR function, which is not computable by a single linear projection due to the required non-linear decision boundary. Figure 2 shows the constructed projection, and its data and parameter vectors along with their set-centric interpretation. In empirical experiments in which we train a Tversky projection to learn XOR with gradient descent under various hyperparameter conditions, we find that: Some initializations of prototypes and features lead to convergence failure; Initializing those parameters from a uniform distribution leads to higher convergence probability compared to normal and orthogonal initialization; Normalizing prototype and object vectors deteriorates convergence probability; *product* and *substractmatch* work best as values of the *intersection reduction* and *difference reduction* hyperparameters; Convergence probability doesn't increase monotonically with the feature bank size; and finally, Tversky projection can model XOR with as little as one feature. See Appendix C.

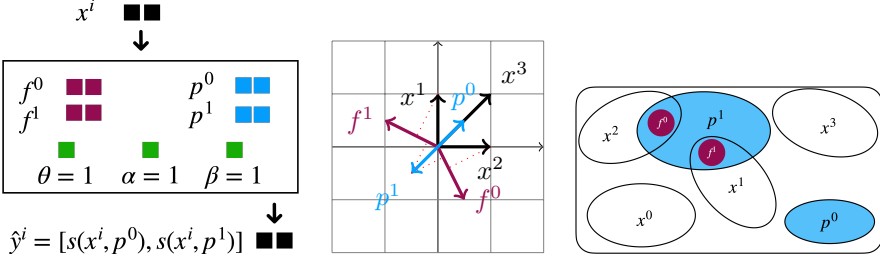

Figure 2: **(left)** A Tversky projection layer with 11 learnable parameters that computes the XOR function. The input is one of the four 2-digit binary numbers encoded as $x^i \in \mathbb{R}^2$. The output vector $\hat{y}^i = [\hat{y}_0^i, \hat{y}_1^i]$ contains the Tversky similarity of $x^i$ to prototypes $p^0$ and $p^1$ computed using features $f^0$ and $f^1$, and coefficients $\theta$, $\alpha$ and $\beta$, such that $\hat{y}_1^i > \hat{y}_0^i \iff xor(x^i) = 1$. **(middle)** Vector representation of data $x^i$ (with $x^0$ not marked in the figure but at the origin), prototypes $p^i$, and features $f^i$. Notice positive dot-products: $x^1 \cdot f^1$, $x^2 \cdot f^0$, $p^1 \cdot f^0$ and $p^1 \cdot f^1$. **(right)** Set representation and Tversky feature matching. Objects are represented as the subset of features with which they have a positive dot product: $x^0 = \{\}$, $x^1 = \{f^1\}$, $x^2 = \{f^0\}$, $x^3 = \{\}$, $p^0 = \{\}$, $p^1 = \{f^0, f^1\}$. $x^0$ and $x^3$ are more similar to $p^0$ than to $p^1$, which has two distinctive features penalizing similarity; thus $xor([0,0]) = 0$ and $xor([1,1]) = 0$. $x^1$ and $x^2$ are more similar to $p^1$, with which they each share one feature; thus $xor([0,1]) = 1$ and $xor([1,0]) = 1$.

### 4.2 LANGUAGE MODELING WITH TVERSKYGPT-2

Table 1: Perplexity (PPL) and parameter counts (Params) of GPT2 and TverskyGPT2 trained for the language modeling task on the PTB dataset. Perplexities are reported on the held out test split.

| Model | Init | Tied Proto | Features | Params | PPL | $\Delta Params$ | $\Delta PPL$ |
|---|---|---|---|---|---|---|---|
| baseline | finetuned | N | | 163 037 184 | 30.52 | | |
| tversky-head | finetuned | N | 16 384 | 175 620 099 | **28.33** | +7.7% | -7.2% |
| baseline | finetuned | Y | | 124 439 808 | **18.31** | | |
| tversky-head | finetuned | Y | 32 768 | 149 605 635 | 18.62 | +20.2% | +1.7% |
| baseline | scratch | N | | 163 037 184 | 111.79 | | |
| tversky-all-1layer | scratch | N | 4 096 | 116 591 655 | **98.22** | -28.5 % | -12.1 % |
| baseline | scratch | Y | | 124 439 808 | 112.81 | | |
| tversky-all-1layer | scratch | Y | 8 192 | 81 140 007 | **103.99** | -34.8% | -7.8 % |

### 4.2.1 Task and Data

We compare the baseline GPT-2 small (Radford et al., 2019) model with its Tversky variants on the Penn Treebank (PTB) language modeling benchmark (Marcus et al., 1993).

### 4.2.2 Method

The following TverskyGPT-2 variants were considered: *tversky-head*, which replaces GPT-2's language modeling head's stack of linear layers with a Tversky projection layer, and *tversky-all*, which also replaces GPT-2's intermediate attention blocks' output projections with Tversky projections. Tversky feature sharing is employed across all Tversky projection layers. For all models, we experiment with prototype tying (tied vs not tied) and initialization (random vs OpenAI's released weights) while training for 50 epochs on PTB's training set, and validating on PTB validation set. The baseline models, and best Tversky models are evaluated on PTB's held-out test set.

### 4.2.3 Results

Tversky language models matched or surpassed baseline perplexity in all settings except the one in which pre-trained weights and prototype tying are employed (Table 1). In that setting, Tversky prototypes and features are still randomly initialized, while the baseline fully connected layers are initialized from the pretrained token embedding matrix. The perplexity gap between the best Tversky neural network and baseline was higher in the tversky-all configuration trained from scratch, with a 7.8% reduction in perplexity when prototypes are tied to input embeddings, and a 12.1% reduction when they are not. In the tied prototype setting, the best Tversky model also has 34.8% fewer parameters. See Appendix B for validation results.

## 4.3 Image Recognition with TverskyResNet-50

### 4.3.1 Task and Data

We experiment with replacing the final linear projection layer in ResNet-50 (He et al., 2016) with a Tversky projection and report the accuracy of the baseline and Tversky variants on the MNIST (LeCun et al., 1998) handwritten digit classification task and the NABirds (Van Horn et al., 2015) bird species classification task.

### 4.3.2 Method

We compare the baseline ResNet-50 to TverskyResNet-50, which replaces the fully connected layer with a Tversky projection layer. Both models are evaluated under different training conditions. In the *Pretrained Backbone* setting, convolutional weights are initialized from ImageNet; otherwise they are randomly initialized. In the *Frozen Backbone* setting, backbone parameters are fixed and only the final projection module is optimized. Tversky prototype and feature banks are randomly initialized in all cases, and multiple feature bank sizes are explored. Because MNIST and NABirds do not provide official three-way splits, we randomly sample 10% of the training data to construct a validation set and reserve the official test splits for final evaluation. Each configuration is trained with multiple random seeds, affecting parameter initialization and train/validation partitioning. Model selection is performed using mean validation accuracy across seeds, and the selected configuration is evaluated on the held-out standard test set. Experimental details, including hyperparameter grids, optimization settings, data preprocessing, augmentation, model selection protocol, and final evaluation results are provided in Appendix Sections E and F.

### 4.3.3 Results

Classification accuracies reported in Table 2 show that Tversky vision models can match or surpass baseline accuracy under the specified experimental conditions. The exception is the frozen backbone setting on NABirds, where the baseline marginally outperforms TverskyResNet-50. In particular, when trained from scratch, TverskyResNet-50 improves test accuracy by 2.36 percentage points.

In this experiment, TverskyResNet-50 has more parameters than the baseline because, in addition to the prototype bank (analogous to the fully connected layer weights), it also includes a feature

Table 2: Accuracy of ResNet-50 (Baseline) and TverskyResNet-50 (Tversky) on the tasks of MNIST handwritten digit classification and NABirds bird species classification. *Pretrained* (True when weights are initialized from ResNet-50/ImageNet, False when they are randomly initialized). *Frozen* (True when only the final projection layer is finetuned, False when the entire model is finetuned)

| Pretrained | Frozen | MNIST | | NABirds | |
|---|---|---|---|---|---|
| | | Tversky | Baseline | Tversky | Baseline |
| False | False | $\mathbf{99.56 \pm 0.06}$ | $99.54 \pm 0.04$ | $\mathbf{65.20 \pm 0.26}$ | $62.84 \pm 0.45$ |
| True | False | $\mathbf{99.60 \pm 0.05}$ | $99.56 \pm 0.04$ | $\mathbf{82.96 \pm 0.07}$ | $82.37 \pm 0.25$ |
| True | True | $\mathbf{86.66 \pm 0.17}$ | $86.64 \pm 0.14$ | $38.73 \pm 0.20$ | $\mathbf{40.25 \pm 0.28}$ |

bank. While the increased parameter count of TverskyResNet-50 could contribute to the observed accuracy improvement, our results in Section 4.2 show that the use of Tversky neural networks can simultaneously result in decreased parameter count and increased performance compared to baseline.

### 4.4 QUALITATIVE ANALYSIS OF TVERSKY NEURAL NETWORKS

Here we present results showing the interpretability of Tversky neural networks. We discuss the explainability of learned prototypes, the quantification of salience, and the algebraic specification of semantic fields enabled by Tversky neural networks.

#### 4.4.1 METHOD

**Salience Measure:** To assess whether salience, as computed using Equation 2, is intelligible to humans, we rank MNIST and NABirds examples by their salience score.

**Prototype Visualization:** To visually compare the prototypes learned by Tversky Neural Networks and classical neural networks, we employ the prototype visualization method described in Section 3.6 and illustrated in Appendix A. This method is used to display the handwritten digit prototypes learned by VisualMNISTNet and VisualTverskyMNISTNet, two neural networks inspired by LeNet-5 (LeCun et al., 1998). Further details are provided in Appendix D.

**Semantic Fields:** Tversky Neural Networks' representation of objects as sets of features allows the algebraic specification of interpretable *semantic fields*. A semantic field is a subset of features resulting from set expressions that are interpretable by humans. For instance, $A \cap B - C$ describes the semantic field capturing the common features of $A$ and $B$ that are absent from $C$. To visualize a semantic field $F$, we retrieve examples $x_i$ with the highest scores in that semantic field: $s = \sum_{f_k \in F} f_k \cdot x_i$.

#### 4.4.2 RESULTS

**Salience:** Figure 1 shows MNIST and NABirds examples ranked from low to high salience according to Equation 2. Higher-ranked stimuli exhibit more object features and greater *informational content*, *familiarity*, and *goodness of form*, consistent with Tversky (1977)'s theory.

**Prototypes:** Figure 3 shows that Tversky prototypes exhibit stroke patterns similar to human handwriting, such as lines and curves, more clearly than those learned by linear projection layers, which exhibit texture patterns that are difficult to interpret. This lack of interpretability, even in this simple domain without background textures, represents a significant limitation of prior approaches. Tversky prototypes are also more *salient* than individual instances: they exhibit more features and appear to combine the features of all possible instances of their class. For example, the prototype for the handwritten "7" has both an upper-left serif and a middle horizontal stroke. The prototypes for the digit "1" and "9" include vertical lines slanted to the right and to the left. The prototype for class "9" combines both open and closed top loops. Figure 16 shows the evolution of learned prototypes over training epochs for both VisualMNISTNet and TverskyVisualMNISTNet. We observed $\alpha > \beta$ in several, but not all, trained Tversky neural networks (see Appendix F.4), indicating that they weigh the distinctive features of individual instances more heavily than those of prototypes when assessing the similarity of instances to prototypes, which is in line with Tversky's theory.

**Visualization of Projection Parameters in Data-Domain**

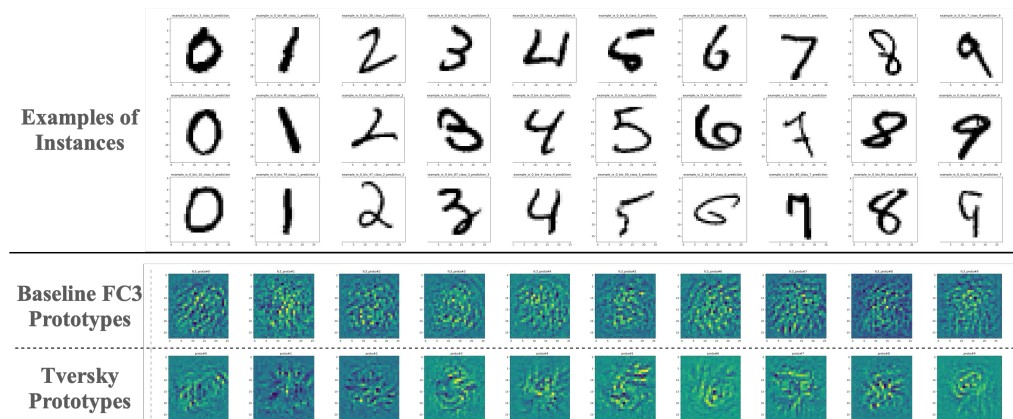

Figure 3: Visualization of prototypes using our input domain projection parameter specification method. Top: 3 examples for each class. Bottom: 10 columns of the final linear projection layer of VisualMNISTNet, and 10 prototype vectors of the Tversky Projection Layer of TverskyMNISTNet. Both models achieve 99% accuracy on the validation set. Handwritten digit features are more perceptible to humans in Tversky prototypes compared to the linear projection prototypes.

**Semantic Fields:** Tversky Neural Networks' set-based representation enables the algebraic specification of semantic fields, which we illustrate in Figure 4 across three modalities. **Lexical stimuli** (A:*love*, B:*like*): $A \cap B$ captures shared evaluative sentiment; $A - B$ captures romance and unconditional love; and $B - A$ captures similarity and comparison. **Handwritten digits:** Common features of two eights retrieve other eights; distinctive features capture a six-like lower stroke and an upper arch prominent in twos, respectively. **Bird species:** Common features of a Least and a Semipalmated Sandpiper retrieve both species; distinctive features retrieve species sharing relevant visual traits (yellow vs. black legs, drooped vs. straight bill). Table 3 and Appendix B.4, B.5 show that semantic fields spanning *adjective degree* (positive, comparative, superlative), word *senses* (*plant*, *sanction*), *morphology* (past tense, plural, progressive), *metonymy* (*White House*, *Pentagon*), *analogy* (*king*:*man*::*?*:*woman*), *sentiment*, *polarity* and *antonymy* are all expressible as concise, human-readable algebraic set expressions over TverskyGPT-2 token representations.

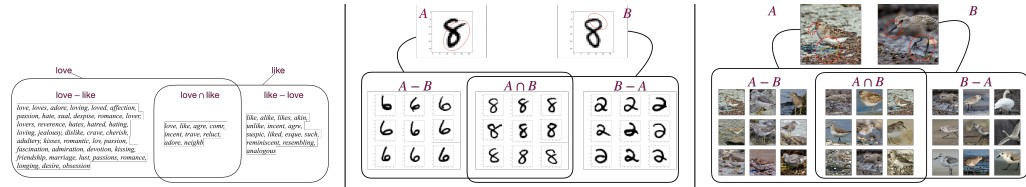

Figure 4: Semantic fields defined as common ($A \cap B$) and distinctive ($A - B, B - A$) features of lexical and visual stimuli. **left:** TverskyGPT-2 tokens *love* and *like*. Distinctively, *love* is associated with romance and unconditional love, while *like* is associated with similarity and comparison. Their common features capture shared evaluative sentiment. **middle:** Two handwritten eights (A) and (B) from MNIST. Their common features retrieve other eights similar to both. A's distinctive features capture a six-like lower stroke, while B's capture an upper right arch, also prominent in twos. **right:** A Least Sandpiper (A) and a Semipalmated Sandpiper (B) from NABirds. Their common features retrieve other sandpipers. A's distinctive features capture its yellow legs and fine drooped bill, while B's capture its black legs and blunt straight black bill.

## 5 DISCUSSION

**Hyperparameter Tuning:** Our experiments suggest the existence of an optimal feature bank size for Tversky projection layers for a given learning task, though determining this value *a priori*

Table 3: Semantic concepts, their algebraic specification using set operations on TverskyGPT-2 tokens, and top tokens ranked by semantic score in the resulting semantic field. Tversky neural networks enable the algebraic specification of interpretable semantic concepts. (Also see Tables 8–12)

| Concept | Semantic Expression | Top-scoring tokens |
|---------|---------------------|--------------------|
| Adjective | $bad \cap good - worse - better - worst - best$ | *lousy, evil, crappy, poor, shitty, nice, terrible, horrible, decent, mediocre, valid, great, excellent* |
| Comparative | $worse \cap better - bad - good - worst - best$ | *sharper, nicer, clearer, smarter, wiser, smoother, safer, hotter, happier, tighter, louder, preferable* |
| Superlative | $worst \cap best - worse - better - bad - good$ | *happiest, safest, quickest, deadliest, finest, hottest, busiest, toughest, darkest, fastest, brightest, holiest* |
| Industrial Plant | $plant - aceae - vegetation - vegetable - planting - flower - herb - vine - crop - tree - mushroom - plantation - leaves$ | *facility, stall, brate, factory, Laboratories, Shed, implant, Berm, Sew, Manufacturing, reactor, Diesel, refinery, strain, distribut, planting, Industrial, microbial, actory, reactors, Unit, Install, depot, spill, Cutter, Indust, Bot, Nuclear, ineries* |

remains an open question. We encourage practitioners to apply standard hyperparameter tuning to further improve performance, especially since Tversky networks are a novel architecture, whereas the baselines we compare against have benefited from years of community-driven refinement.

**Convergence:** Under particular conditions, some features or prototypes may remain permanently inactive and receive zero gradient throughout training. This phenomenon is analogous to the "dead expert" issue in MoEs (Dai et al., 2024). We illustrated this issue in the XOR domain, which contains only four data points (see Figure 13, bottom row). However, we did not observe this issue in our language modeling experiments, or our image recognition models which showed consistent convergence over several random seeds (Appendix E and F).

**Interpretation of Weight Tying in Language Models:** Our experiments with GPT-2 on PTB showed that weight tying can be beneficial in language modeling heads employing both Tversky and linear projections. Our interpretation of projection layers as measuring similarity between stimuli and prototypes explains the adequacy of weight tying in language models: token embeddings are prototypes, and the language modeling head predicts output tokens by measuring the similarity of output representations to these prototypes.

## CONCLUSION

This work introduces a differentiable similarity function based on Tversky's theory of psychological similarity, and derives basic neural network building blocks, specifically the Tversky similarity and projection layers, offering an alternative to the geometric similarity functions that underlie most modern neural architectures. Our experiments show that Tversky projection layers can improve over linear projection layers in performance and parameter efficiency in image recognition and language modeling tasks. Tversky's set-based representation also enables the algebraic specification of *semantic fields*, which are human-interpretable feature subsets that support fine-grained analysis of learned representations in any modality. For instance, querying the distinctive features of a Least Sandpiper and a Semipalmated Sandpiper retrieves birds with yellow legs and drooped bill, and birds with black legs and straight black bill, respectively. We also propose a novel data-domain visualization method enabling interpretation of learned prototypes and features directly in input space at any network depth. Taken together, these contributions suggest a shift in the similarity paradigm underlying deep learning: from geometric comparison toward explicit feature-matching grounded in an established theory of psychological similarity.

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

## A  ILLUSTRATION OF DATA-DOMAIN VISUALIZATION OF PROJECTION LAYERS

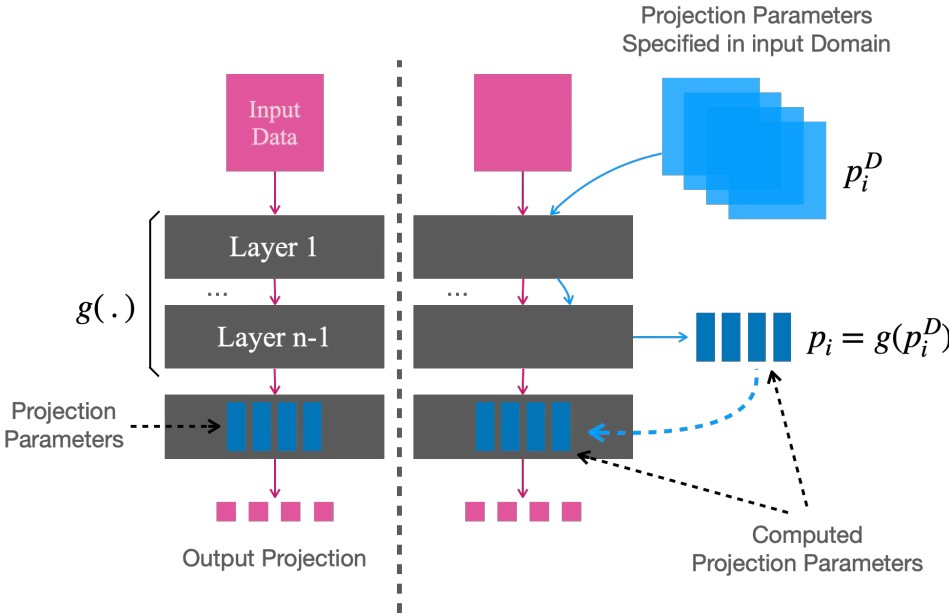

Figure 5: Illustration of our proposed data-domain visualization of projection parameters. **Left:** Classical deep neural network with a deep projection layer (Layer n) and its projection parameters illustrated as blue vectors. **Right:** Our proposed method. Projection parameters are specified as tensors of the same shape as the input data $(p_i^D)$, and forwarded through the neural network up to layer n-1 (the $g(.)$ sub-network). The obtained vectors $p_i = g(p_i^D)$ are used as projection parameters in Layer n. Using this method, the effective neural network parameter count is larger because $p_i^D$ is typically larger than $p_i$. However, this method enables visualizing the projection parameters in the input domain.

# B  EXPERIMENT 006: [TVERSKY]GPT-2 LANGUAGE MODEL ON PTB

## B.1  INITIALIZATION: OPENAI RELEASED WEIGHTS

| | | | | | Validation Perplexity | | | | | | | | | |
|---|---|---|---|---|---|---|---|---|---|---|---|---|---|---|
| | | | difference → | | | | ignorematch | | | | | substractmatch | | |
| | | | intersection → | | gmean | max | mean | min | product | gmean | max | mean | min | product |
| Init | model | tie-proto | features | params | PPL | PPL | PPL | PPL | PPL | PPL | PPL | PPL | PPL | PPL |
| finetuned | baseline | N | | 163037184 | 34.85 | 34.85 | 34.85 | **34.85** | 34.85 | **34.85** | 34.85 | **34.85** | **34.85** | **34.85** |
| finetuned | tversky-head | N | 1024 | 163823619 | 52.42 | 70.11 | 51.80 | 57.66 | 76.73 | 61.59 | 51.90 | 58.15 | 85.37 | 120.22 |
| finetuned | tversky-head | N | 2048 | 164610051 | 40.92 | 54.44 | 41.23 | 46.79 | 53.00 | 48.71 | 41.28 | 47.91 | 66.95 | 86.26 |
| finetuned | tversky-head | N | 4096 | 166182915 | 34.47 | 38.52 | 34.44 | 40.32 | 40.32 | 42.82 | 34.54 | 41.89 | 64.80 | 57.10 |
| finetuned | tversky-head | N | 8192 | 169328643 | **32.56** | **32.80** | **32.53** | 38.80 | 34.50 | 38.89 | **32.49** | 39.09 | 61.66 | 48.72 |
| finetuned | tversky-head | N | 12288 | 172474371 | 32.84 | 32.92 | 32.82 | 40.82 | 32.32 | 41.18 | 32.81 | 41.29 | 70.24 | 52.26 |
| finetuned | tversky-head | N | 16384 | 175620099 | 33.62 | 36.33 | 33.62 | 39.75 | *32.31 | 40.71 | 33.63 | 40.67 | 77.18 | 52.32 |
| finetuned | tversky-head | N | 32768 | 188203011 | 38.36 | 40.34 | 38.98 | 49.35 | 33.94 | 49.48 | 38.79 | 47.17 | 109.10 | 62.37 |
| finetuned | tversky-all-1layer | N | 1024 | 114232359 | 145.50 | 208.61 | 142.82 | 154.14 | 219.63 | 172.13 | 144.59 | 155.65 | 638.28 | 297.61 |
| finetuned | tversky-all-1layer | N | 2048 | 115018791 | 118.23 | 172.70 | 115.15 | 120.06 | 166.49 | 121.00 | 114.88 | 119.46 | 257.80 | 253.91 |
| finetuned | tversky-all-1layer | N | 4096 | 116591655 | 95.65 | 143.67 | 95.51 | 121.13 | 124.50 | 122.75 | 95.79 | 121.51 | 157.91 | 203.54 |
| finetuned | tversky-all-1layer | N | 8192 | 119737383 | 91.40 | 94.86 | 91.44 | 93.09 | 99.75 | 93.14 | 91.45 | 93.53 | 149.20 | 142.79 |
| finetuned | tversky-all-1layer | N | 12288 | 122883111 | 90.39 | 94.49 | 90.79 | 92.54 | 94.00 | 92.98 | 90.74 | 92.40 | 147.16 | 133.41 |
| finetuned | tversky-all-1layer | N | 16384 | 126028839 | 90.51 | 96.41 | 91.22 | 95.39 | 91.08 | 95.64 | 91.23 | 95.06 | 156.30 | - |
| finetuned | tversky-all-1layer | N | 32768 | 138611751 | 96.00 | 101.08 | 96.44 | 100.78 | 92.68 | - | - | - | - | - |

Table 4: Validation perplexity comparison with **init=finetuned** and **tie-proto=N**.

| | | | | | Validation Perplexity | | | | | | | | | |
|---|---|---|---|---|---|---|---|---|---|---|---|---|---|---|
| | | | difference → | | | | ignorematch | | | | | substractmatch | | |
| | | | intersection → | | gmean | max | mean | min | product | gmean | max | mean | min | product |
| Init | model | tie-proto | features | params | PPL | PPL | PPL | PPL | PPL | PPL | PPL | PPL | PPL | PPL |
| finetuned | baseline | Y | | 124439808 | *19.99 | *19.99 | *19.99 | *19.99 | *19.99 | *19.99 | *19.99 | *19.99 | *19.99 | *19.99 |
| finetuned | tversky-head | Y | 1024 | 125226243 | 90.22 | 39.28 | 86.30 | 61.22 | 69.68 | 61.29 | 87.64 | 61.08 | 81.07 | 94.78 |
| finetuned | tversky-head | Y | 2048 | 126012675 | 59.95 | 28.69 | 56.18 | 67.96 | 43.19 | 71.83 | 57.38 | 68.19 | 46.20 | 58.08 |
| finetuned | tversky-head | Y | 4096 | 127585539 | 23.26 | 27.15 | 24.01 | 24.54 | 28.91 | 23.37 | 24.08 | 23.22 | 30.84 | 35.15 |
| finetuned | tversky-head | Y | 8192 | 130731267 | 21.17 | 22.31 | 21.25 | 21.44 | 22.37 | 21.43 | 21.23 | 21.50 | 22.03 | 23.24 |
| finetuned | tversky-head | Y | 12288 | 133876995 | 21.07 | 22.64 | 21.09 | 21.33 | 20.99 | 21.32 | 21.08 | 21.35 | 21.98 | 21.81 |
| finetuned | tversky-head | Y | 16384 | 137022723 | 20.86 | 21.86 | 20.85 | 21.12 | 20.77 | 21.14 | 20.85 | 21.12 | 22.88 | - |
| finetuned | tversky-head | Y | 32768 | 149605635 | 20.70 | 21.33 | 20.68 | 20.82 | 20.46 | - | - | - | - | - |
| finetuned | tversky-all-1layer | Y | 1024 | 75634983 | 134.68 | 188.07 | 132.79 | 174.71 | 212.09 | 166.07 | 132.21 | 174.75 | 741.72 | 321.28 |
| finetuned | tversky-all-1layer | Y | 2048 | 76421415 | 101.62 | 182.26 | 102.64 | 131.28 | 164.13 | 134.14 | 103.46 | 130.69 | 235.55 | 228.58 |
| finetuned | tversky-all-1layer | Y | 4096 | 77994279 | 77.13 | 87.87 | 76.87 | 96.56 | 109.65 | 97.91 | 76.94 | 100.72 | 198.39 | 190.20 |
| finetuned | tversky-all-1layer | Y | 8192 | 81140007 | 70.07 | 88.16 | 69.75 | 80.32 | 83.06 | 79.63 | 69.71 | 79.65 | 97.48 | 117.50 |
| finetuned | tversky-all-1layer | Y | 12288 | 84285735 | 67.61 | 79.07 | 66.50 | 76.51 | 73.95 | 76.66 | 66.51 | 75.85 | 101.56 | 97.24 |
| finetuned | tversky-all-1layer | Y | 16384 | 87431463 | 69.54 | 79.76 | 69.22 | 78.32 | 73.03 | 79.33 | 69.22 | 78.85 | 100.12 | - |
| finetuned | tversky-all-1layer | Y | 32768 | 100014375 | 73.33 | 99.82 | 73.12 | 84.67 | 67.19 | - | - | - | - | - |

Table 5: Validation perplexity comparison with **init=finetuned** and **tie-proto=Y**.

## B.2 INITIALIZATION: RANDOM WEIGHTS

| | | | difference → | | ignorematch | | | | | substractmatch | | | | |
|---|---|---|---|---|---|---|---|---|---|---|---|---|---|---|
| | | | intersection → | | gmean | max | mean | min | product | gmean | max | mean | min | product |
| Init | model | tie-proto | features | params | PPL | PPL | PPL | PPL | PPL | PPL | PPL | PPL | PPL | PPL |
| scratch | baseline | N | | 163037184 | 134.06 | 134.06 | 134.06 | 134.06 | 134.06 | 134.06 | 134.06 | 134.06 | **134.06** | 134.06 |
| scratch | tversky-head | N | 1024 | 163823619 | 146.74 | 218.66 | 146.10 | 157.30 | 184.36 | 155.74 | 145.65 | 158.21 | 496.50 | 168.50 |
| scratch | tversky-head | N | 2048 | 164610051 | 147.92 | 192.12 | 148.02 | 145.72 | 171.54 | 155.64 | 147.64 | 145.66 | 320.49 | 140.48 |
| scratch | tversky-head | N | 4096 | 166182915 | 168.12 | 161.04 | 167.65 | 176.67 | 162.97 | 157.44 | 167.05 | 155.93 | 147.83 | **133.47** |
| scratch | tversky-head | N | 8192 | 169328643 | 210.03 | 204.28 | 217.75 | 182.50 | 173.63 | 182.08 | 215.71 | 182.47 | 205.87 | 154.21 |
| scratch | tversky-head | N | 12288 | 172474371 | 227.69 | 178.52 | 241.49 | 212.82 | 186.98 | 212.26 | 236.39 | 212.04 | 203.83 | 163.90 |
| scratch | tversky-head | N | 16384 | 175620099 | 229.84 | 202.26 | 250.62 | 228.47 | 213.86 | 231.28 | 248.62 | 231.49 | 228.31 | 200.96 |
| scratch | tversky-head | N | 32768 | 188203011 | 182.45 | 162.13 | 190.04 | 190.92 | 252.01 | 198.72 | 187.12 | 197.99 | 223.08 | 218.13 |
| scratch | tversky-all-1layer | N | 1024 | 114232359 | nan | 173.25 | 126.67 | 138.92 | 169.06 | nan | 127.19 | 139.09 | 271.09 | 347.79 |
| scratch | tversky-all-1layer | N | 2048 | 115018791 | 121.06 | 135.85 | 120.29 | 123.94 | 130.72 | 126.68 | 120.58 | 124.17 | 163.99 | 249.04 |
| scratch | tversky-all-1layer | N | 4096 | 116591655 | **120.89** | **127.62** | 121.11 | 123.97 | *117.59 | 124.16 | 121.26 | **123.86** | 136.27 | 225.91 |
| scratch | tversky-all-1layer | N | 8192 | 119737383 | 128.32 | 134.91 | 129.87 | 126.38 | 118.06 | 126.72 | 129.45 | 126.31 | 155.86 | 140.68 |
| scratch | tversky-all-1layer | N | 12288 | 122883111 | 130.54 | 131.29 | 132.46 | 127.68 | 125.03 | 127.69 | 132.42 | 127.72 | 159.30 | 137.27 |
| scratch | tversky-all-1layer | N | 16384 | 126028839 | 129.24 | 131.03 | 131.82 | 128.37 | 132.29 | 128.33 | 131.94 | 128.01 | 163.26 | - |
| scratch | tversky-all-1layer | N | 32768 | 138611751 | 129.11 | 132.33 | 130.26 | 129.51 | 136.46 | - | - | - | - | - |

Table 6: Validation perplexity comparison with **init=scratch** and **tie-proto=N**.

| | | | difference → | | ignorematch | | | | | substractmatch | | | | |
|---|---|---|---|---|---|---|---|---|---|---|---|---|---|---|
| | | | intersection → | | gmean | max | mean | min | product | gmean | max | mean | min | product |
| Init | model | tie-proto | features | params | PPL | PPL | PPL | PPL | PPL | PPL | PPL | PPL | PPL | PPL |
| scratch | baseline | Y | | 124439808 | 136.04 | **136.04** | 136.04 | 136.04 | 136.04 | 136.04 | 136.04 | 136.04 | **136.04** | **136.04** |
| scratch | tversky-head | Y | 1024 | 125226243 | 149.22 | 213.67 | 148.81 | 155.98 | 193.72 | 153.13 | 149.07 | 156.29 | 360.26 | 271.20 |
| scratch | tversky-head | Y | 2048 | 126012675 | 147.91 | 179.89 | 148.12 | 145.80 | 169.92 | 155.16 | 147.83 | 145.83 | 175.95 | 154.26 |
| scratch | tversky-head | Y | 4096 | 127585539 | 167.44 | 168.53 | 168.55 | 150.11 | 161.85 | 151.59 | 168.02 | 149.89 | 152.11 | 145.52 |
| scratch | tversky-head | Y | 8192 | 130731267 | 197.06 | 207.89 | 204.79 | 181.21 | 168.29 | 179.76 | 201.95 | 180.99 | 183.96 | 156.37 |
| scratch | tversky-head | Y | 12288 | 133876995 | 212.50 | 211.17 | 223.91 | 202.83 | 179.87 | 204.57 | 221.02 | 202.63 | 214.04 | 165.64 |
| scratch | tversky-head | Y | 16384 | 137022723 | 210.90 | 152.18 | 227.04 | 217.16 | 204.58 | 214.09 | 226.02 | 216.78 | 238.05 | - |
| scratch | tversky-head | Y | 32768 | 149605635 | 162.49 | 151.23 | 171.88 | 176.74 | 216.54 | - | - | - | - | - |
| scratch | tversky-all-1layer | Y | 1024 | 75634983 | 142.00 | 174.76 | 138.77 | 152.44 | 182.22 | 154.00 | 138.65 | 152.31 | 250.44 | 389.16 |
| scratch | tversky-all-1layer | Y | 2048 | 76421415 | 132.41 | 141.77 | 130.55 | 136.88 | 146.27 | 144.29 | 130.69 | 137.51 | 180.40 | 332.45 |
| scratch | tversky-all-1layer | Y | 4096 | 77994279 | **129.46** | 137.46 | **129.64** | 130.57 | 128.84 | 130.26 | **129.66** | 130.61 | 146.93 | 175.75 |
| scratch | tversky-all-1layer | Y | 8192 | 81140007 | 136.73 | 145.34 | 139.16 | *125.86 | 128.05 | 125.98 | 139.13 | 125.96 | 151.85 | 151.07 |
| scratch | tversky-all-1layer | Y | 12288 | 84285735 | 140.86 | 156.21 | 144.50 | 128.38 | 134.98 | 127.85 | 144.61 | 128.45 | 156.44 | 140.26 |
| scratch | tversky-all-1layer | Y | 16384 | 87431463 | 143.59 | 160.09 | 148.72 | 130.69 | 140.38 | 129.88 | 148.77 | 130.50 | 153.77 | - |
| scratch | tversky-all-1layer | Y | 32768 | 100014375 | 154.05 | - | - | 140.73 | 150.33 | - | - | - | - | - |

Table 7: Validation perplexity comparison with **init=scratch** and **tie-proto=Y**.

## B.3 COMPUTATION TIMES AND RESOURCES

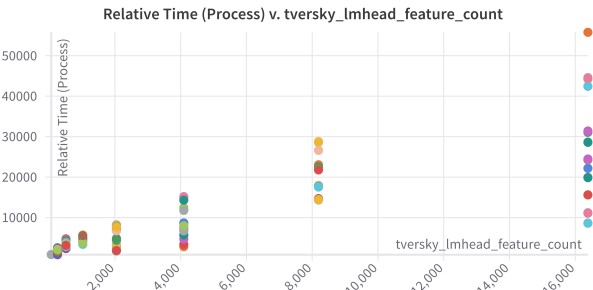

Figure 6: Sample of wall-clock times in seconds for TverskyGPT2 experiments on PTB as a function of the feature bank size. 2 GPUs were used for each run. GPU models varied. They included NVIDIA A100-SXM4-40GB, NVIDIA H100 80GB HBM3, and NVIDIA RTX 6000 Ada Generation.

### B.4 VISUALIZATION OF SEMANTIC FIELDS IN TVERSKYGPT2'S SET-CENTRIC TOKEN REPRESENTATION

Figures 7 ,8 ,9 ,10 and 11 show examples of visualizations of semantic fields formed using set expressions on TverskyGPT2 tokens represented as sets. For each token, the top 1000 features (sorted by dot-product with the token) are considered, and represented as the smallest colored circles. Tokens are represented as yellow circles, and connected to the features they comprise with grey lines. All distinct intersections of features are colored in the same color, and connected to a circle of the same color representing a distinctive semantic field. Semantic fields of interest are annotated with a text box listing the top 50 tokens by semantic score (See Section 4.4) in that semantic field, permitting its visualization. A force-directed graph layout is employed to position the circles corresponding to tokens, features, and semantic fields. Note that the Ġ character (U+0120) represents space in GPT2 tokenization, distinguishing tokens that appear at the beginning of words from other tokens.

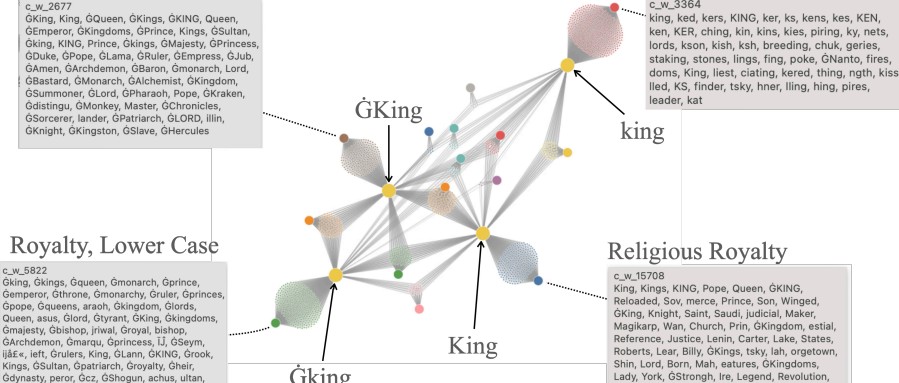

Figure 7: Visualization of the semantic fields formed by the distinctive features of related tokens. Notice that TverskyGPT2 tokens capture semantics related to orthographic markings, and morphological function. The $king$ token (as in baking) is related to other suffixes such as $ked$, $ker$ and $kes$ (as in baked, baker and bakes)
.

TverskyGPT2 Tokens: Polysemy

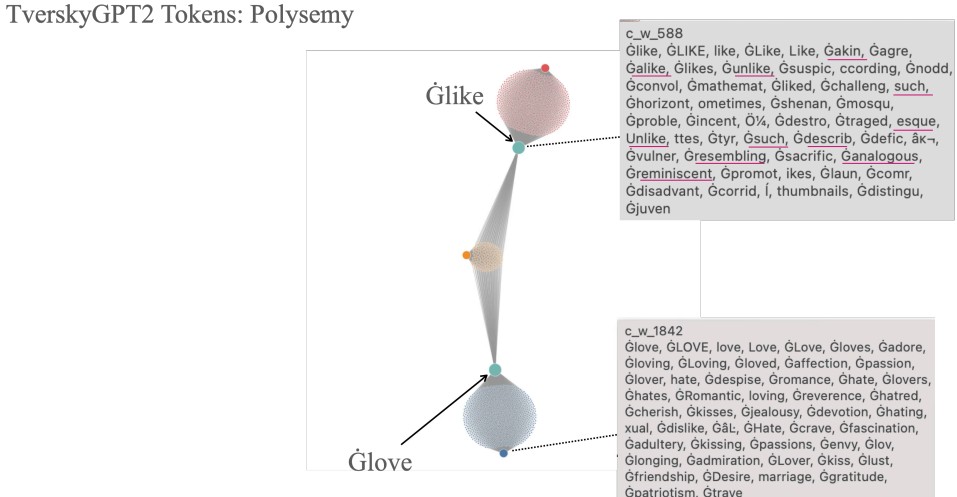

Figure 8: Visualization of the semantic fields formed by the distinctive features of tokens *like* (*like − love*) and *love* (*love − like*). *like*, distinctively from *love*, which also captures the sense of sentiment, captures the sense of similarity, with tokens such as *akin*, *alike*, *unlike* and *reminiscent*.

TverskyGPT2 Tokens: Verb Forms

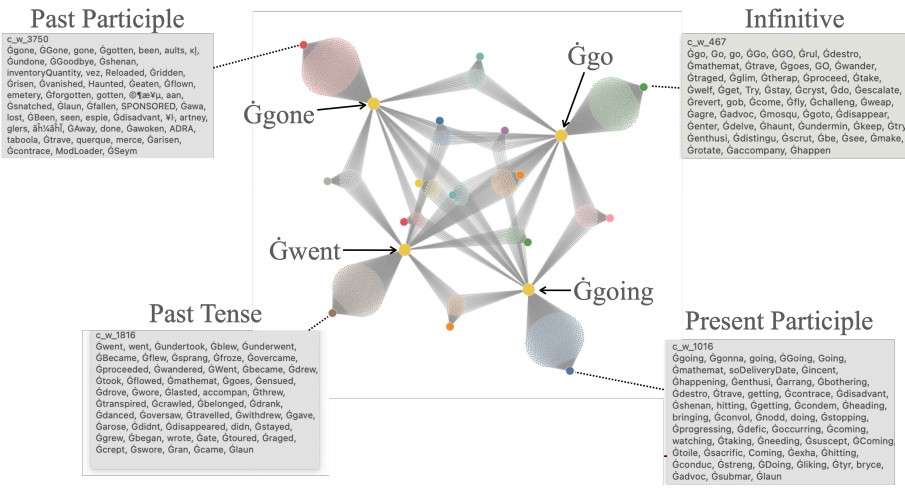

Figure 9: Visualization of the semantic fields formed by the distinctive features of tokens *go* (*go − gone − went − going*), *gone* (*gone − go − went − going*), *went* (*went − gone − go − going*), and *going* (*going − gone − went − go*), respectively specifying the concept of verb forms *infinitive*, *past participle*, *past tense* and *present participle*.

TverskyGPT2 Tokens: Degrees of Comparison of Adjectives

**Adjective**

c_w_2089_w_922
Ġbad, Ġgood, bad, good, ĠGOOD, Good, ĠGood, ĠBAD, Bad, ĠBad, Ġlousy, Evil, Ġrul, Ġcrappy, Ġnodd, Ġtherap, Ġcondem, Poor, Ġshitty, cellent, Ġevil, Ġnewsp, Ġnice, Ġterrible, Ġpoor, Ġnotor, Excellent, Ġencour, ĠEvil, âģ*, Ġnaughty, Ġenthusi, Ġfortun, ĠExcellent, Ġingred, Ġsugg, Ġdecent, Ġrotten, Ġmisunder, Ġtrave, Ġmediocre, Ġthous, Ġunbeliev, Ġhorrible, Valid, Ġconfir, Ġgreat, Ġexcellent, Ġsuspic, xIJ

**Superlative**

c_w_1266_w_5290
Ġworst, Ġbest, worst, best, ĠBEST, ĠWorst, Ġhappiest, Ġsafest, Ġquickest, Best, Ġdeadliest, ĠBest, Ġfinest, Ġbusiest, Ġtoughest, Ġhottest, Ġdarkest, Ġfastest, Ġhardest, Ġgreatest, Ġeasiest, Ġbrightest, Ġdeepest, Ġlongest, Ġcoolest, Ġstrongest, Ġcheapest, Ġheaviest, Ġweakest, highest, Ġclosest, orst, Ġpoorest, Ġwidest, Ġsmartest, ottest, Ġshortest, Ġtallest, Ġlowest, Ġbiggest, Ġoldest, Ġhighest, largest, Ġhars, same, Ġsimplest, iest, Ġholiest, Ġinfamous, Ġwealthiest

**Comparative**

c_w_1365_w_4785
Ġworse, better, Ġbetter, ĠWorse, Better, Ġsharper, Ġnicer, Ġclearer, Ġsmarter, Ġwiser, Ġsmoother, ĠBetter, Ġsafer, Ġhotter, Ġhappier, Ġquicker, Ġtighter, Ġstronger, Ġpoorer, Ġlouder, Downloadha, Ġhealthier, Ġharsher, Ġcheaper, Ġworsened, arser, Ġtaller, Ġbigger, Ġpreferable, Ġcleaner, Ġbrighter, Ġharder, Ġsimpler, Ġricher, ĠFaster, Ġfiner, oother, Ġwealthier, Ġthicker, Ġtougher, Ġweaker, Ġquieter, Ġstricter, raq, ĠBET, Ġeasier, Ġworsen, ModLoader, Ġfaster, Ġnarrower

Ġbad  Ġbest  Ġgood  Ġworst  Ġworse  Ġbetter

Figure 10: Visualization of the semantic fields formed by the distinctive features of the common features of tokens $bad$ and $good$ ($bad \cap good - worse - better - worst - best$), $worse$ and $better$ ($worse \cap better - bad - good - worst - best$), and $worst$ and $best$ ($worst \cap best - bad - good - worse - better$). These semantic fields algebraically specify the concepts of *adjective*, *comparative*, and *superlative*.

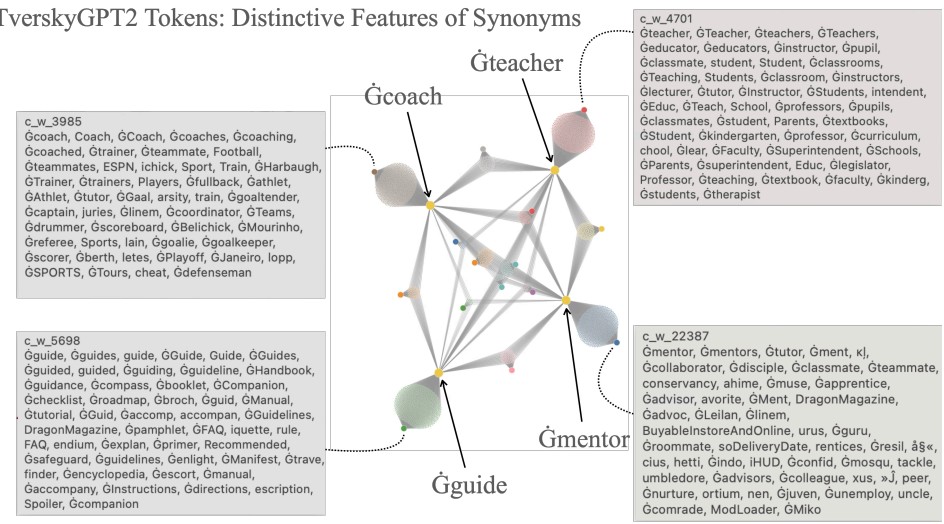

TverskyGPT2 Tokens: Distinctive Features of Synonyms

Ġteacher  Ġcoach

c_w_4701
Ġteacher, ĠTeacher, Ġteachers, ĠTeachers, Ġeducator, Ġeducators, Ġinstructor, Ġpupil, Ġclassmate, student, Student, Ġclassrooms, ĠTeaching, Students, Ġclassroom, Ġinstructors, Ġlecturer, Ġtutor, ĠInstructor, ĠStudents, intendent, ĠEduc, ĠTeach, School, Ġprofessors, Ġpupils, Ġclassmates, Ġstudent, Parents, Ġtextbooks, ĠStudent, Ġkindergarten, Ġprofessor, Ġcurriculum, chool, Ġlear, ĠFaculty, ĠSuperintendent, ĠSchools, ĠParents, Ġsuperintendent, Educ, Ġlegislator, Professor, Ġteaching, Ġtextbook, Ġfaculty, Ġkinderg, Ġstudents, Ġtherapist

c_w_3985
Ġcoach, Coach, ĠCoach, Ġcoaches, Ġcoaching, Ġcoached, Ġtrainer, Ġteammate, Football, Ġteammates, ESPN, ichick, Sport, Train, ĠHarbaugh, ĠTrainer, Ġtrainers, Players, Ġfullback, Ġathlet, ĠAthlet, Ġtutor, ĠGaal, arsity, train, Ġgoaltender, Ġcaptain, juries, Ġlinem, Ġcoordinator, ĠTeams, Ġdrummer, Ġscoreboard, ĠBelichick, ĠMourinho, Ġreferee, Sports, lain, Ġgoalie, Ġgoalkeeper, Ġscorer, Ġberth, letes, ĠPlayoff, ĠJaneiro, lopp, ĠSPORTS, ĠTours, cheat, Ġdefenseman

c_w_5698
Ġguide, Ġguides, guide, ĠGuide, Guide, ĠGuides, Ġguided, guided, Ġguiding, Ġguideline, ĠHandbook, Ġguidance, Ġcompass, Ġbooklet, ĠCompanion, Ġchecklist, Ġroadmap, Ġbroch, Ġguid, ĠManual, Ġtutorial, ĠGuid, Ġaccomp, accompan, ĠGuidelines, DragonMagazine, Ġpamphlet, ĠFAQ, iquette, rule, FAQ, endium, Ġexplan, Ġprimer, Recommended, Ġsafeguard, Ġguidelines, Ġenlight, ĠManifest, Ġtrave, finder, Ġencyclopedia, Ġescort, Ġmanual, Ġaccompany, ĠInstructions, Ġdirections, escription, Spoiler, Ġcompanion

c_w_22387
Ġmentor, Ġmentors, Ġtutor, Ġment, кļ, Ġcollaborator, Ġdisciple, Ġclassmate, Ġteammate, conservancy, ahime, Ġmuse, Ġapprentice, Ġadvisor, avorite, ĠMent, DragonMagazine, Ġadvoc, ĠLeilan, Ġlinem, BuyableInstoreAndOnline, urus, Ġguru, Ġroommate, soDeliveryDate, rentices, Ġresil, â§«, cius, hetti, Ġindo, iHUD, Ġconfid, Ġmosqu, tackle, umbledore, Ġadvisors, Ġcolleague, xus, »Ĵ, peer, Ġnurture, ortium, nen, Ġjuven, Ġunemploy, uncle, Ġcomrade, ModLoader, ĠMiko

Ġmentor  Ġguide

Figure 11: Visualization of the semantic fields capturing the distinctive senses of close synonyms $coach$, $teacher$, $mentor$, and $guide$. Notice that the distinctive features of $coach$ ($coach - teacher - mentor - guide$) shows terms related to sport such as *trainer*, *teammate*, *Football* and *ESPN* whereas the distinctive features of $teacher$ ($teacher - coach - mentor - guide$) show terms related to formal education such as *educator*, *classroom*, *instructor*, *lecturer* and *professor*.

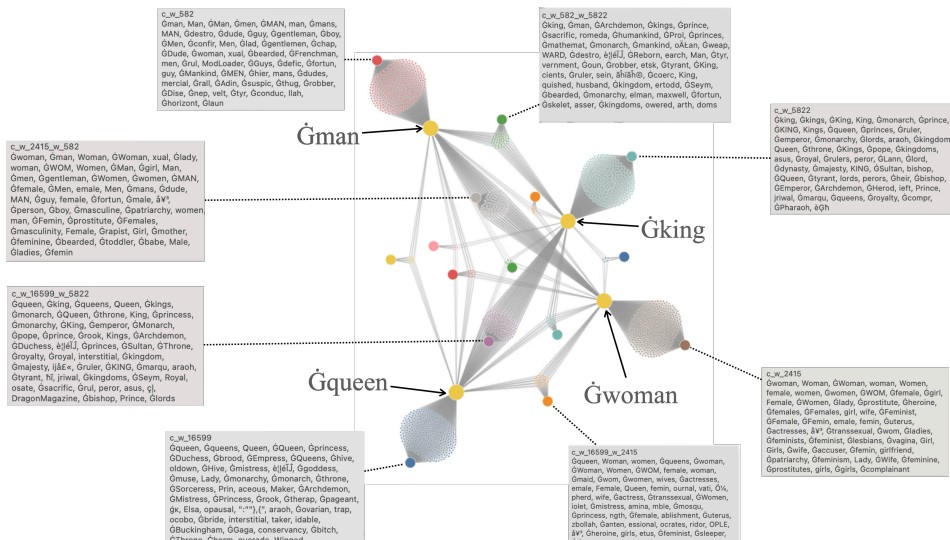

Figure 12: Visualization of the semantic fields capturing the distinctive feature of tokens *man*, *woman*, *king* and *queen*, and the distinctive semantics of the common features of *man* and *king*, *man* and *woman*, *woman* and *queen*, *queen* and *king*, and *man* and *woman*

## B.5 ALGEBRAIC LEXICAL SEMANTIC EXPRESSIONS OVER TVERSKYGPT2'S TOKENS

Tables 8–12 present semantic fields spanning a broad taxonomy of concept types, algebraically specified as set expressions over TverskyGPT-2 token representations.

Table 8: Examples of semantic fields TverskyGPT-2. Part 1/5: Morphology and Lexical sense

| Category | Concept & Expression | Top-scoring tokens |
|---|---|---|
| Morphology | Past tense
$ran \cap walked - run - walk - running - walking$ | walked, ran, wandered, crawled, Became, smelled, stumbled, persisted, danced, prevailed, sprang, resided, risked, exited, sailed, overcame, froze, toured, apologized, leapt |
| | Progressive
$running \cap walking - ran - walked - run - walk$ | walking, running, running, Running, Running, wandering, walking, crawling, Walking, marching, barking, rocking, whining, campaigning, kicking, stumbling, driving, hiking, touring, hopping |
| | Plural noun
$dogs \cap cats - dog - cat$ | cats, dogs, puppies, kittens, Dogs, dogs, pets, tigers, pige, Cats, chimpanzees, lions, goats, cats, Animals, Horses, reptiles, wolves, dolphins, wolves |
| | Negation prefix
$(unlikely - likely) \cap (unfair - fair) \cap (unusual - usual)$ | improbable, misunder, unfair, unlikely, suspic, unethical, unusual, unsus, Seym, ModLoader, unexpected, uthor, unforeseen, theless, aution, unorthodox, untled, icult, cffff, unnatural |
| Lexical sense | Like (comparison)
$unlike \cap like - dislike$ | unlike, like, LIKE, Like, like, Like, Unlike, Unlike, ccording, akin, reminiscent, agre, incent, whereas, eatures, ecause, Seym, nodd, sacrific, tery |
| | Like (Evaluation)
$like \cap dislike - unlike$ | dislike, LIKE, disliked, liking, like, like, reluct, mathemat, Like, Like, despise, disdain, suspic, blat, agre, comr, hating, disapprove, disadvant, ometimes |
| | Sanction (permit)
$sanction - punishments - Prosecut - disapproval - shaming - disband$ | sanction, sanctioned, sanctions, enance, sponsorship, sponsoring, spons, Grant, ction, displayText, legitim, , č, , sponsor, BCC, authorizing, decree, patronage, awarding |
| | Sanction (punish)
$sanction - approve - permit - allow - authorize$ | sanction, sanctioned, sanctions, punishments, displayText, , Prosecut, discipl, disadvant, disapproval, sovere, č, condem, shaming, reperc, tsky, disband, ENDED, enance, retribution |

Table 9: Examples of semantic fields TverskyGPT-2. Part 2/5: Metonymy and Analogy

| Category | Concept & Expression | Top-scoring tokens |
|---|---|---|
| Metonymy | White House (government) $White \cap House - building - home - residence - architecture$ | House, White, White, House, WHITE, Senate, Houses, Senate, HOUSE, Oval, Congressional, Speaker, artisan, Democrats, Capitol, Commons, congressional, Republican, Democratic, Congress |
| | Hollywood (film industry) $Hollywood - California - city - neighborhood - suburb$ | Hollywood, ollywood, Film, Spielberg, Actor, Movie, movie, Movies, Movie, Disney, Celeb, MGM, filmmaking, cinematic, TMZ, Kubrick, Cannes, filmmakers, Oscars, actresses |
| | Pentagon (military) $Pentagon - shape - geometry - building - architecture$ | Pentagon, DOD, Military, Mattis, USAF, Lockheed, Kremlin, Guant, neocons, Defense, CIA, psey, CLASSIFIED, anamo, Petraeus, FBI, Defense, Clapper, Army, Kissinger |
| Analogy | king:man :: ?:woman $(king - man) \cap woman$ | king, Woman, queen, Queen, Duchess, princess, woman, Woman, Queen, Lady, ournal, woman, queens, women, goddess, female, Women, foundland, kings, heroine |
| | Cairo:Egypt :: ?:China $(Cairo - Egypt) \cap China$ | Cairo, Beijing, China, Shanghai, China, Chinese, ijing, Chinatown, Manila, stanbul, Sina, Riyadh, Beirut, Xin, ongyang, Havana, Guang, Islamabad, Moscow, atown |
| | Cairo:Egypt :: ?:Brazil $(Cairo - Egypt) \cap Brazil$ | Brazil, Cairo, Brazil, Janeiro, Rio, Brazilian, Sao, querque, Spain, Lisbon, Uruguay, Buenos, Paulo, iago, razil, Paris, Mexico, Bras, Urug, Manila |
| | actor:man :: ?:woman $(actor - man) \cap woman$ | actress, actresses, Actress, actor, Actor, Woman, Actor, woman, heroine, prostitute, vati, comedian, dancer, filmmaker, actors, emale, woman, contrace, xual, girlfriend |
| | doctor:hospital :: ?:school $(doctor - hospital) \cap school$ | doctor, doctor, school, school, School, teacher, psychiatrist, dentist, physician, veterinarian, educator, mathemat, SCHOOL, Doctors, soDeliveryDate, chool, Teacher, tutor, Doctors, teachers |

Table 10: Examples of semantic fields TverskyGPT-2. Part 3/5: Domain, Relation and Sentiment

| Category | Concept & Expression | Top-scoring tokens |
|---|---|---|
| Domain | Academic title
$professor \cap doctor - student - hospital - medicine$ | professor, doctor, doctor, psychiatrist, Professor, lecturer, Professor, physicist, Doctor, scientist, iologist, professors, physician, veterinarian, rabbi, mathematician, dentist, psychologist, Doctor, biologist |
| | Legal language
$court \cap law - judge - sports - game - play - field$ | court, law, law, Court, Court, Law, Law, COURT, courts, court, LAW, laws, Courts, statute, Lawyers, Justice, courthouse, Attorney, Legal, lawy |
| | Programming
$Python \cap Java - snake - coffee - animal - drink$ | Python, Java, Python, Java, Django, python, ython, Clojure, python, Scala, PHP, Lua, Âű, Drupal, Lua, java, Jakarta, Unix, Lisp, Haskell |
| Relation | Sibling
$brother \cap sister - father - mother - son - daughter - family$ | sister, brother, sisters, sibling, brother, Brother, Sister, brothers, Brother, cousin, siblings, cousins, Sisters, comrade, niece, brethren, Brothers, nephew, neighbour, friend |
| | Spouse
$husband \cap wife - man - woman - partner - family - child$ | husband, wife, Wife, husband, husbands, wife, wives, spouse, fian, spouses, fiance, bride, Mrs, widow, wives, marital, Bride, granddaughter, niece, boyfriend |
| Sentiment | Negative emotion
$anger \cap sadness - happy - joy - fear - anxiety$ | sadness, anger, BuyableInstoreAndOnline, Anger, terness, fury, bitterness, sorrow, indignation, oneliness, resentment, cynicism, mourning, disgust, loneliness, engeance, remorse, grief, saddened, anguish |
| | Mild positive
$good \cap nice - great - excellent - perfect - best - wonderful$ | nice, good, nice, good, Good, Nice, Good, Nice, nicer, GOOD, bad, nifty, romeda, sane, pleasant, decent, Friendly, osuke, positives, shitty |
| | Surprise
$surprise \cap shock - fear - anger - sadness - disgust$ | shock, surprise, shock, shocks, surprises, Surprise, Shock, Shock, aston, surpr, shocked, surprised, surprising, shocking, bombshell, aback, miracle, stunned, dazz, startled |

Table 11: Examples of semantic fields TverskyGPT-2. Part 4/5: Polarity

| Category | Concept & Expression | Top-scoring tokens |
|---|---|---|
| Polarity | Bravery (positive)
$(brave \cup afraid) \cap (good - bad \cup positive - negative)$ | brave, courageous, isSpecialOrderable, mathemat, trave, advoc, misunder, suspic, streng, challeng, comr, toget, enthusi, bravery, psychiat, valiant, dilig, successfully, Honest, redes |
| | Bravery (negative)
$(brave \cup afraid) \cap (bad - good \cup negative - positive)$ | afraid, destro, ModLoader, ashamed, fearful, disadvant, defic, misunder, BAD, reluct, isSpecialOrderable, scared, psychiat, suspic, Fear, terrified, inventoryQuantity, trave, frightened, bad |
| | Usefulness (positive)
$(useful \cup useless) \cap (good - bad)$ | useful, good, good, Useful, GOOD, worthwhile, helpful, Good, usefulness, cellent, mathemat, fruitful, valuable, challeng, indispensable, invaluable, ngth, Excellent, useless, usable |
| | Usefulness (negative)
$(useful \cup useless) \cap (bad - good)$ | useless, bad, BAD, disadvant, worthless, bad, ineffective, unus, unemploy, defic, Bad, pointless, lousy, Worse, faulty, superflu, undesirable, counterproductive, destro, Evil |
| | Temperature (cold)
$(cold \oplus hot) \cap (small - big \cup low - high)$ | cold, cold, Cold, Low, chilly, LOW, colder, low, Cold, misunder, Small, nodd, small, Low, small, shallow, Small, defic, horizont, low |
| | Temperature (hot)
$(cold \oplus hot) \cap (big - small \cup high - low)$ | hot, mathemat, hotter, Hot, big, HOT, Hot, cryst, hot, HUGE, hots, Big, dirty, contrace, cold, misunder, hots, hottest, confir, BIG |
| | Wealth
$rich \oplus poor$ | poor, rich, poor, Poor, Poor, richer, poorer, Rich, defic, rich, wealthier, impoverished, richest, wealthy, Rich, poorest, needy, affluent, mathemat, wealthiest |
| | Antiquity
$ancient \oplus modern$ | modern, Modern, Ancient, cients, ancient, antiquity, modern, Ancient, prehistoric, medieval, Antiqu, ieval, archaic, Modern, historic, Medieval, mathemat, contemporary, indo, millennia |

Table 12: Examples of semantic fields TverskyGPT-2. Part 5/5: Antonymy. While *unhappy* appears in the semantic field of *happy* (highlighted in red), set difference isolates the distinct senses of each pole: $happy - unhappy$ and $unhappy - happy$.

| Category | Concept & Expression | Top-scoring tokens |
|---|---|---|
| Antonymy | (semantic field of happy) 
 *happy* | happy, happy, Happy, happiest, happier, Happy, pleased, unhappy, ecstatic, thrilled, satisfied, joyful, delighted, glad, happiness, grateful, thankful, Happiness, happily, cheerful, amused, joy, proud, merry, jealous, Honest, reluct, rejoice, honoured, triumphant |
| | (semantic field of unhappy) 
 *unhappy* | unhappy, dissatisfied, discont, untled, dissatisf, disple, dissatisfaction, happy, happiest, disgruntled, unsatisf, reluct, happier, discontent, displeasure, miserable, unlucky, Seym, disillusion, uneasy, appiness, distraught, undesirable, Rebell, contrace, unemploy, disadvant, suspic, annoyed, irritated |
| | (happy in contrast to unhappy) 
 $happy - unhappy$ | happy, happy, Happy, Happy, happiest, happier, pleased, thrilled, ecstatic, glad, delighted, satisfied, joyful, grateful, thankful, proud, happily, happiness, amused, cheerful, Happiness, joy, Merry, honoured, smiles, Honest, stoked, pleasant, rejoice, merry |
| | (unhappy in contrast to happy) 
 $unhappy - happy$ | unhappy, dissatisfied, untled, dissatisf, discont, disple, dissatisfaction, unsatisf, disgruntled, displeasure, discontent, reluct, unlucky, disillusion, undesirable, unemploy, disadvant, miserable, contrace, unwelcome, Seym, intolerable, unworthy, defic, distraught, demoral, irritated, Rebell, uneasy, estranged |

## C  EXPERIMENT 007: TVERSKYXORNET ON XOR

### C.1  TASK AND DATA

In this experiment, we train a single Tversky projection layer to learn the XOR function via gradient descent under various hyper-parameter conditions. Figure 13 shows that some initializations of Tversky projection layers may lead to gradient descent optimization failure. This experiment empirically analyses its sensitivity to hyperparameters when trained to model the XOR function.

### C.2  METHOD

To empirically estimate the sensitivity of Tversky projection layer's convergence to its hyperparameters, we train 12 960 xor models consisting of the following combinations of hyperparameters:

- 6 intersection reduction methods {*min*, *max*, *product*, *mean*, *gmean*, *softmin*};
- 2 difference reduction methods {*ignorematch*, *substractmatch*};
- 2 normalization modes: {false, true};
- 6 feature counts: {1, 2, 4, 8, 16, 32};
- 3 prototype initialization distributions {uniform, normal, orthogonal};
- 3 feature bank initialization distributions {uniform, normal, and orthogonal };
- 9 random seeds.

Each model is trained for 1000 epochs. Tables 13, 14, 15, and 16 reports the average and standard error of the convergence indicator (whether accuracy is 100%) of the trained models marginalized by various combinations of hyperparameters. We refer to this variable as convergence probability $p(conv)$ in our results.

### C.3  RESULTS

**Initialization of prototypes and features**  Initializing both features and prototypes by sampling from the uniform distribution resulted in the highest convergence probability (Table 14).

**Reduction of measures of intersections and differences**  Using *product* and *substractmatch* resulted in the highest convergence probability. (Table 13)

**Normalization**  In this experiment, normalizing prototype and object vectors prior to calculating Tversky similarity decreased the convergence probability. (Table 15)

**Feature count**  Tversky projection layer successfully modeled *xor* with as little as 1 feature. $p(conv)$ was maximal with 16 features, but not did monotonously increase with feature count.

While linear projection layers cannot learn non-linear decision boundaries without composition with non-linear *activation* functions, Figure 14 shows that a single Tversky projection layers can learn complex non-linear decision boundaries even in low-dimensional vector space.

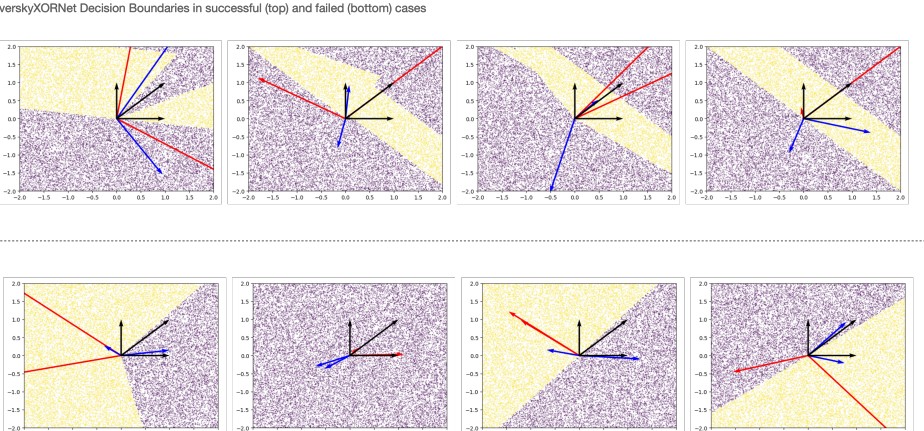

Figure 13: Example of learned TverskyProjection layers optimized to model the xor function. While multiple solutions are possible (top row), some initializations do not lead to convergence (bottom row). The decision boundaries were drawn as follows. Random input vectors [x,y] are uniformly sampled in the range [-2,-2] to [2,2] the tip of each vector is ploted as a colored dot, with the color representing the trained Tversky Projection layer's output. Tversky projection layers modeling the $xor$ function, which has a boolean domain and range extend $xor$ to the real vector space. Successful models should show data points [0,0] and [1,1] in purple (class 0), and [0,1] and [1,0] in yellow (class 1)

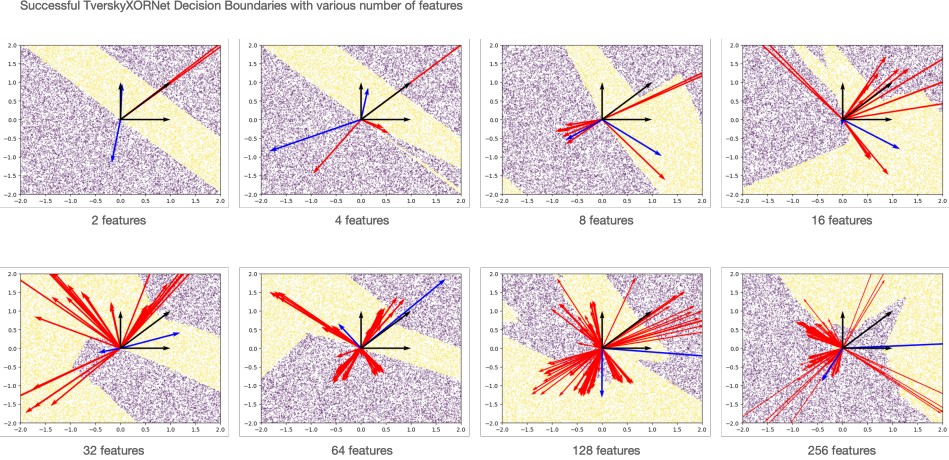

Figure 14: Example of successful Tversky decision boundaries modeling the XOR function with 2, 4, 8, 16, 32, 64, 128 and 256 features. The presence of clusters of features are frequent in overparameterized models.

Table 13: Mean ± std error of loss and accuracy for TverskyXORNet with 6 intersection reduction methods ($A \cap B$) and 2 difference reduction methods ($A - B$). Each row corresponds to the aggregate of 3x3x2x6x9 = 972 training sessions with: 3 fbank and prototype initialization methods(orthogonal, normal, uniform), 2 normalization modes (normalized, and not normalized), 6 feature bank sizes (1, 2, 4, 8, 16, 32), and 9 random seeds. In each case, models were trained for 1000 epochs

| $A \cap B$ | $A - B$ | n | loss | acc | best acc | p(conv) |
|---|---|---|---|---|---|---|
| product | substractmatch | 972 | $0.27 \pm 0.01$ | $0.83 \pm 0.01$ | 1.0 | $0.53 \pm 0.02$ |
| mean | substractmatch | 972 | $0.28 \pm 0.01$ | $0.82 \pm 0.01$ | 1.0 | $0.51 \pm 0.02$ |
| max | ignorematch | 972 | $0.35 \pm 0.01$ | $0.78 \pm 0.01$ | 1.0 | $0.47 \pm 0.02$ |
| max | substractmatch | 972 | $0.32 \pm 0.01$ | $0.80 \pm 0.01$ | 1.0 | $0.44 \pm 0.02$ |
| softmin | substractmatch | 972 | $0.35 \pm 0.01$ | $0.80 \pm 0.01$ | 1.0 | $0.42 \pm 0.02$ |
| min | ignorematch | 972 | $0.31 \pm 0.01$ | $0.78 \pm 0.01$ | 1.0 | $0.42 \pm 0.02$ |
| softmin | ignorematch | 972 | $0.34 \pm 0.01$ | $0.78 \pm 0.01$ | 1.0 | $0.38 \pm 0.02$ |
| mean | ignorematch | 972 | $0.42 \pm 0.01$ | $0.73 \pm 0.01$ | 1.0 | $0.26 \pm 0.01$ |
| product | ignorematch | 972 | $0.41 \pm 0.01$ | $0.75 \pm 0.01$ | 1.0 | $0.23 \pm 0.01$ |
| min | substractmatch | 972 | $0.50 \pm 0.00$ | $0.68 \pm 0.00$ | 1.0 | $0.02 \pm 0.00$ |
| gmean | ignorematch | 972 | $nan \pm nan$ | $0.50 \pm 0.00$ | 0.5 | $0.00 \pm 0.00$ |
| gmean | substractmatch | 972 | $nan \pm nan$ | $0.50 \pm 0.00$ | 0.5 | $0.00 \pm 0.00$ |

Table 14: Mean ± std error of loss and accuracy for TverskyXORNet with 3 fbank and prototype initialization methods (*fbank init*, and *proto init*). Each row corresponds to the aggreate of 6x2x2x6x9 = 1,296 training sessions with: 6 intersection reduction methods (min, max, product, mean, gmean, softmin) and 2 difference reduction methods (ignorematch, substractmatch), 2 normalization modes (normalized, and not normalized), 6 feature bank sizes (1, 2, 4, 8, 16, 32), and 9 random seeds. In each case, models were trained for 1000 epochs.

| fbank init | proto init | n | loss | acc | best acc | p(conv) |
|---|---|---|---|---|---|---|
| uniform | uniform | 1296 | $0.27 \pm 0.01$ | $0.78 \pm 0.01$ | 1.0 | $0.41 \pm 0.01$ |
| uniform | normal | 1296 | $0.36 \pm 0.01$ | $0.72 \pm 0.01$ | 1.0 | $0.34 \pm 0.01$ |
| normal | uniform | 1296 | $0.34 \pm 0.01$ | $0.74 \pm 0.01$ | 1.0 | $0.32 \pm 0.01$ |
| normal | normal | 1296 | $0.35 \pm 0.01$ | $0.74 \pm 0.01$ | 1.0 | $0.31 \pm 0.01$ |
| uniform | orthogonal | 1296 | $0.39 \pm 0.01$ | $0.70 \pm 0.01$ | 1.0 | $0.30 \pm 0.01$ |
| orthogonal | uniform | 1296 | $0.37 \pm 0.01$ | $0.72 \pm 0.01$ | 1.0 | $0.29 \pm 0.01$ |
| normal | orthogonal | 1296 | $0.37 \pm 0.01$ | $0.73 \pm 0.01$ | 1.0 | $0.28 \pm 0.01$ |
| orthogonal | normal | 1296 | $0.37 \pm 0.01$ | $0.73 \pm 0.01$ | 1.0 | $0.26 \pm 0.01$ |
| orthogonal | orthogonal | 1296 | $0.39 \pm 0.01$ | $0.71 \pm 0.01$ | 1.0 | $0.24 \pm 0.01$ |

Table 15: Mean ± std error of loss and accuracy for TverskyXORNet with 2 normalization modes (normalized, and not normalized). Each row corresponds to the aggreate of 3x3x6x2x6x9 = 5832 training sessions with: 3 fbank and prototype initialization methods(orthogonal, normal, uniform). 6 intersection reduction methods (min, max, product, mean, gmean, softmin) and 2 difference reduction methods (ignorematch, substractmatch), 6 feature bank sizes (1, 2, 4, 8, 16, 32), and 9 random seeds. In each case, models were trained for 1000 epochs.

| normalize | n | loss | acc | best acc | p(conv) |
|---|---|---|---|---|---|
| False | 5832 | $0.33 \pm 0.00$ | $0.74 \pm 0.00$ | 1.0 | $0.34 \pm 0.01$ |
| True | 5832 | $0.38 \pm 0.00$ | $0.72 \pm 0.00$ | 1.0 | $0.27 \pm 0.01$ |

Table 16: Mean $\pm$ std error of loss and accuracy for TverskyXORNet with 6 feature bank sizes (1, 2, 4, 8, 16, 32). Each row corresponds to the aggreate of 3x3x6x2x2x9 = 1944 training sessions with: 3 fbank and prototype initialization methods(orthogonal, normal, uniform). 6 intersection reduction methods (min, max, product, mean, gmean, softmin) and 2 difference reduction methods (ignorematch, substractmatch), 2 normalization modes (normalized, and not normalized), and 9 random seeds. In each case, models were trained for 1000 epochs.

| fbank size | n | loss | acc | best acc | p(conv) |
|---|---|---|---|---|---|
| 16.0 | 1944 | $0.25 \pm 0.01$ | $0.79 \pm 0.00$ | 1.0 | $0.42 \pm 0.01$ |
| 8.0 | 1944 | $0.28 \pm 0.01$ | $0.78 \pm 0.00$ | 1.0 | $0.39 \pm 0.01$ |
| 4.0 | 1944 | $0.30 \pm 0.01$ | $0.76 \pm 0.00$ | 1.0 | $0.38 \pm 0.01$ |
| 32.0 | 1944 | $0.31 \pm 0.01$ | $0.76 \pm 0.00$ | 1.0 | $0.33 \pm 0.01$ |
| 2.0 | 1944 | $0.45 \pm 0.01$ | $0.67 \pm 0.00$ | 1.0 | $0.20 \pm 0.01$ |
| 1.0 | 1944 | $0.54 \pm 0.01$ | $0.61 \pm 0.00$ | 1.0 | $0.12 \pm 0.01$ |

## D  EXPERIMENT 009: [VISUAL][TVERSKY]MNISTNET

Two neural networks, VisualMNISTNet, and VisualTverskyMNISTNet (Figure 15) are trained to perform MNIST handwritten digit classification. The two neural network architectures share the same convolutional feature extraction stack yielding a 36-dimensional vector representation of the input image. They differ in how those vectors are projected onto the 10-dimensional output vectors corresponding to 10 digit classes. VisualMNISTNet employs a stack of 120, 84, and 10 unit linear projection layers following LeCun et al. (1998). However, the ReLU activation function is used instead of the logistic sigmoid. VisualTverskyMNISTNet employs a single Tversky Projection layer with 10 prototypes and 20 features. Both neural networks employ the method described in Section 3.6, facillitating our qualitative analysis of the learned prototypes and features.

Two additional neural networks, MNISTNet and TverskyMNISTNet, which are identical to their *Visual* counterparts, but do not employ our parameter visualization method are also trained to serve as baseline for accuracy. These neural networks have fewer parameters as the projection parameters are specified in their vector form, which is more compact.

MNISTNet is three times smaller than LeNet-5 due to the design of its convolution stack, which outputs a 36-dimensional vector which are the concatenation of three 12-dimensional vectors obtained by averaging 3 convolutional feature maps across their spatial dimensions.

TverskyMNISTNet, with only 7K parameters is also 3 times smaller than MNISTNet because it employs a single Tversky Projection layer instead of a stack linear projection layers and non-linear activation functions. All 4 neural networks are trained for 1000 epochs with a batch size of 500. Dropout is applied to the output of the convolution stack at the rate of 0.05. Adam optimizer is used with learning rate 0.002, and weight decay rate 0.00001.

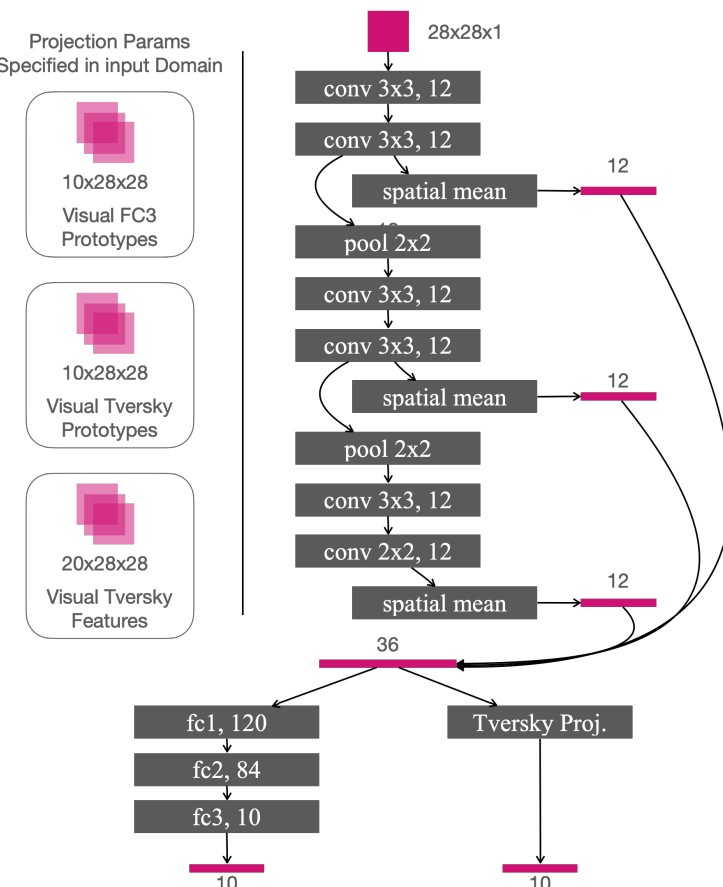

Figure 15: The neural network architecture used in our qualitative analysis experiments.

Table 17: Parameter count and accuracy of baseline and tversky convolutional neural networks trained for MNIST handwritten digit classification. The "visual" variants specify tversky prototypes and features, and the third fully connected layer's parameters in the input space as 28x28 matrices. All models were trained for 1000 epochs.

| Model | Params | Valid ACC |
|---|---|---|
| MNISTNet | 21 394 | 99.1% |
| VisualMNISTNet | 28 394 | 99.1% |
| TverskyMNISTNet | 7 023 | 98.7% |
| VisualTverskyMNISTNet | 29 463 | 98.7% |

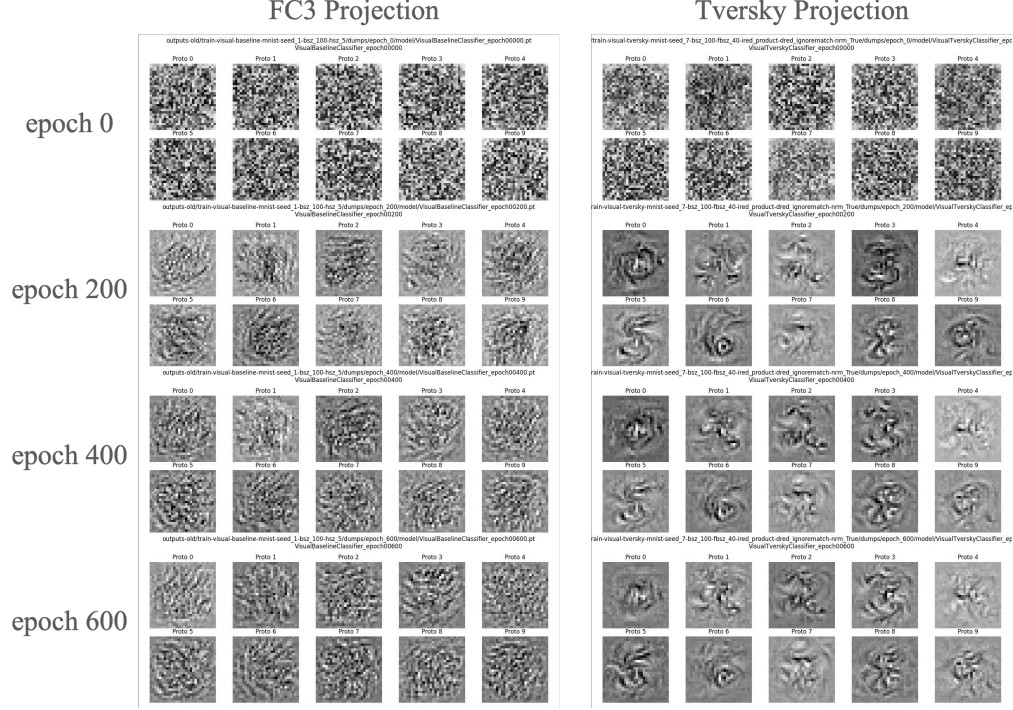

Figure 16: Learned visual prototypes in the baseline and tversky variants of the neural network described in Figure 15 after epochs 0, 200, 400 and 600.

# E  EXPERIMENT 101: [TVERSKY]RESNET-50 ON MNIST

Models cross a variety of hyperparameters (batch size, grad norm clipping threshold) and model initialization (random/image net weights) and param update (update/freeze back one) strategies were trained with 5 random seeds. Each seed corresponds to a different model initial initialization, and a different train/val splits of the training sets (See Section E.1) All models were trained for 200 epochs. The SGD optimizer was applied with learning rate 0.03, momentum 0.9, weight decay 0.0001. MNIST images were also up sampled to $56 \times 56$ input size.

The mean and standard deviation of model accuracies in the train and validation portions are used for model selection. (See Section E.2)

The best Tversky and baseline models in each model initial and parameter update strategy were selected and evaluated on the test set to produce the final mean and standard deviation of model accuracies on the test set. (See Section E.3)

## E.1  HYPERPARAMETER SEARCH

Results are organized by batch size and gradient clipping combinations. **Bold** entries indicate the best performing model within each (Pretrained, Freeze) condition. Accuracies are shown as mean $\pm$ std.

Table 18: Results for Batch Size=32, Gradient Clipping=1

| Pretrained | Freeze | Model | Seeds | Train Acc (%) | Valid Acc (%) |
|---|---|---|---|---|---|
| True | False | baseline | 5 | $100.00 \pm 0.00$ | $99.50 \pm 0.08$ |
| True | False | tversky (fbank=2) | 5 | $64.59 \pm 17.82$ | $63.70 \pm 17.64$ |
| True | False | tversky (fbank=3) | 5 | $76.65 \pm 13.52$ | $76.00 \pm 13.28$ |
| True | False | tversky (fbank=5) | 5 | $96.20 \pm 5.21$ | $95.80 \pm 5.07$ |
| True | False | tversky (fbank=8) | 5 | $98.06 \pm 4.32$ | $97.38 \pm 4.70$ |
| **True** | **False** | **tversky (fbank=13)** | **5** | $100.00 \pm 0.00$ | $99.52 \pm 0.09$ |
| True | False | tversky (fbank=21) | 5 | $100.00 \pm 0.00$ | $99.48 \pm 0.06$ |
| True | False | tversky (fbank=34) | 5 | $100.00 \pm 0.00$ | $99.47 \pm 0.07$ |
| True | False | tversky (fbank=55) | 5 | $100.00 \pm 0.00$ | $99.51 \pm 0.08$ |
| True | True | baseline | 5 | $84.87 \pm 0.13$ | $84.42 \pm 0.30$ |
| True | True | tversky (fbank=2) | 5 | $20.37 \pm 7.09$ | $19.78 \pm 6.78$ |
| True | True | tversky (fbank=3) | 5 | $33.37 \pm 8.99$ | $33.08 \pm 8.97$ |
| True | True | tversky (fbank=5) | 5 | $51.57 \pm 11.78$ | $51.86 \pm 12.02$ |
| True | True | tversky (fbank=8) | 5 | $73.40 \pm 6.94$ | $73.14 \pm 7.11$ |
| True | True | tversky (fbank=13) | 5 | $72.51 \pm 19.63$ | $71.65 \pm 19.82$ |
| True | True | tversky (fbank=21) | 5 | $81.44 \pm 7.65$ | $80.46 \pm 7.62$ |
| True | True | tversky (fbank=34) | 6 | $86.89 \pm 1.48$ | $84.73 \pm 0.82$ |
| **True** | **True** | **tversky (fbank=55)** | **5** | $88.12 \pm 2.47$ | $85.58 \pm 1.91$ |
| False | False | baseline | 5 | $100.00 \pm 0.00$ | $99.52 \pm 0.07$ |
| False | False | tversky (fbank=2) | 5 | $13.95 \pm 5.60$ | $13.69 \pm 5.63$ |
| False | False | tversky (fbank=3) | 5 | $9.89 \pm 0.04$ | $9.68 \pm 0.32$ |
| False | False | tversky (fbank=5) | 5 | $20.45 \pm 7.55$ | $19.94 \pm 7.45$ |
| False | False | tversky (fbank=8) | 5 | $56.46 \pm 31.47$ | $55.95 \pm 31.38$ |
| False | False | tversky (fbank=13) | 5 | $60.64 \pm 25.37$ | $60.23 \pm 25.37$ |
| False | False | tversky (fbank=21) | 5 | $100.00 \pm 0.00$ | $99.47 \pm 0.08$ |
| False | False | tversky (fbank=34) | 5 | $100.00 \pm 0.00$ | $99.48 \pm 0.09$ |
| **False** | **False** | **tversky (fbank=55)** | **5** | $100.00 \pm 0.00$ | $99.54 \pm 0.09$ |

Table 19: Results for Batch Size=32, Gradient Clipping=10

| Pretrained | Freeze | Model | Seeds | Train Acc (%) | Valid Acc (%) |
|---|---|---|---|---|---|
| True | False | baseline | 5 | $100.00 \pm 0.00$ | $99.51 \pm 0.07$ |
| True | False | tversky (fbank=2) | 5 | $45.18 \pm 14.84$ | $44.28 \pm 14.45$ |
| True | False | tversky (fbank=3) | 5 | $60.70 \pm 12.54$ | $60.06 \pm 13.00$ |
| True | False | tversky (fbank=5) | 5 | $86.50 \pm 5.37$ | $86.12 \pm 5.11$ |
| True | False | tversky (fbank=8) | 5 | $97.96 \pm 4.56$ | $97.44 \pm 4.61$ |
| True | False | tversky (fbank=13) | 5 | $100.00 \pm 0.00$ | $99.48 \pm 0.05$ |
| **True** | **False** | **tversky (fbank=21)** | **5** | $100.00 \pm 0.00$ | $99.51 \pm 0.08$ |
| True | False | tversky (fbank=34) | 5 | $100.00 \pm 0.00$ | $99.49 \pm 0.06$ |
| True | False | tversky (fbank=55) | 5 | $100.00 \pm 0.00$ | $99.50 \pm 0.05$ |
| **True** | **True** | **baseline** | **5** | $84.87 \pm 0.14$ | $84.31 \pm 0.37$ |
| True | True | tversky (fbank=2) | 5 | $18.24 \pm 8.47$ | $17.66 \pm 8.29$ |
| True | True | tversky (fbank=3) | 5 | $19.86 \pm 11.48$ | $19.54 \pm 11.60$ |
| True | True | tversky (fbank=5) | 5 | $18.53 \pm 4.83$ | $18.15 \pm 4.72$ |
| True | True | tversky (fbank=8) | 5 | $31.47 \pm 29.29$ | $31.41 \pm 29.62$ |
| True | True | tversky (fbank=13) | 5 | $59.10 \pm 35.04$ | $58.56 \pm 34.78$ |
| True | True | tversky (fbank=21) | 5 | $39.88 \pm 41.09$ | $39.44 \pm 40.59$ |
| True | True | tversky (fbank=34) | 5 | $42.30 \pm 39.70$ | $41.78 \pm 39.19$ |
| True | True | tversky (fbank=55) | 5 | $41.11 \pm 42.76$ | $39.92 \pm 41.30$ |
| **False** | **False** | **baseline** | **5** | $100.00 \pm 0.00$ | $99.47 \pm 0.01$ |
| False | False | tversky (fbank=2) | 5 | $9.89 \pm 0.04$ | $9.68 \pm 0.32$ |
| False | False | tversky (fbank=3) | 5 | $9.89 \pm 0.04$ | $9.68 \pm 0.32$ |
| False | False | tversky (fbank=5) | 5 | $9.89 \pm 0.03$ | $9.68 \pm 0.32$ |
| False | False | tversky (fbank=8) | 5 | $29.84 \pm 30.70$ | $29.59 \pm 30.68$ |
| False | False | tversky (fbank=13) | 5 | $45.66 \pm 40.90$ | $45.12 \pm 40.52$ |
| False | False | tversky (fbank=21) | 5 | $49.55 \pm 46.21$ | $49.18 \pm 46.00$ |
| False | False | tversky (fbank=34) | 5 | $66.46 \pm 46.06$ | $66.23 \pm 45.60$ |
| False | False | tversky (fbank=55) | 5 | $64.07 \pm 49.20$ | $63.66 \pm 48.97$ |

Table 20: Results for Batch Size=256, Gradient Clipping=1

| Pretrained | Freeze | Model | Seeds | Train Acc (%) | Valid Acc (%) |
|---|---|---|---|---|---|
| True | False | baseline | 5 | $100.00 \pm 0.00$ | $99.61 \pm 0.06$ |
| True | False | tversky (fbank=2) | 5 | $41.14 \pm 17.28$ | $40.66 \pm 17.05$ |
| True | False | tversky (fbank=3) | 5 | $48.55 \pm 12.96$ | $48.07 \pm 12.93$ |
| True | False | tversky (fbank=5) | 5 | $88.23 \pm 10.96$ | $87.96 \pm 10.89$ |
| True | False | tversky (fbank=8) | 5 | $94.18 \pm 8.69$ | $93.55 \pm 8.79$ |
| True | False | tversky (fbank=13) | 5 | $100.00 \pm 0.00$ | $99.58 \pm 0.11$ |
| True | False | tversky (fbank=21) | 5 | $100.00 \pm 0.00$ | $99.59 \pm 0.09$ |
| **True** | **False** | **tversky (fbank=34)** | **5** | $100.00 \pm 0.00$ | $99.62 \pm 0.06$ |
| True | False | tversky (fbank=55) | 5 | $100.00 \pm 0.00$ | $99.61 \pm 0.06$ |
| True | True | baseline | 5 | $87.35 \pm 0.15$ | $86.02 \pm 0.56$ |
| True | True | tversky (fbank=2) | 5 | $37.59 \pm 1.61$ | $37.46 \pm 1.42$ |
| True | True | tversky (fbank=3) | 5 | $50.81 \pm 4.25$ | $50.79 \pm 4.15$ |
| True | True | tversky (fbank=5) | 5 | $70.79 \pm 3.38$ | $70.75 \pm 3.77$ |
| True | True | tversky (fbank=8) | 5 | $83.31 \pm 2.68$ | $82.32 \pm 2.57$ |
| True | True | tversky (fbank=13) | 5 | $87.81 \pm 0.25$ | $86.01 \pm 0.42$ |
| True | True | tversky (fbank=21) | 5 | $89.47 \pm 0.43$ | $85.99 \pm 0.65$ |
| True | True | tversky (fbank=34) | 5 | $92.36 \pm 0.13$ | $86.72 \pm 0.18$ |
| **True** | **True** | **tversky (fbank=55)** | **5** | $94.67 \pm 0.17$ | $86.85 \pm 0.20$ |
| False | False | baseline | 5 | $100.00 \pm 0.00$ | $99.34 \pm 0.07$ |
| False | False | tversky (fbank=2) | 5 | $24.69 \pm 11.52$ | $24.19 \pm 11.06$ |
| False | False | tversky (fbank=3) | 5 | $20.42 \pm 12.33$ | $20.13 \pm 12.13$ |
| False | False | tversky (fbank=5) | 5 | $26.43 \pm 19.91$ | $26.31 \pm 19.87$ |
| False | False | tversky (fbank=8) | 5 | $57.04 \pm 31.77$ | $56.33 \pm 31.57$ |
| False | False | tversky (fbank=13) | 5 | $88.32 \pm 12.56$ | $87.73 \pm 12.22$ |
| False | False | tversky (fbank=21) | 5 | $94.26 \pm 8.47$ | $93.69 \pm 8.38$ |
| False | False | tversky (fbank=34) | 5 | $100.00 \pm 0.00$ | $99.33 \pm 0.11$ |
| **False** | **False** | **tversky (fbank=55)** | **5** | $100.00 \pm 0.00$ | $99.34 \pm 0.06$ |

Table 21: Results for Batch Size=256, Gradient Clipping=10

| Pretrained | Freeze | Model | Seeds | Train Acc (%) | Valid Acc (%) |
|---|---|---|---|---|---|
| **True** | **False** | **baseline** | **5** | $100.00 \pm 0.00$ | $99.61 \pm 0.05$ |
| True | False | tversky (fbank=2) | 5 | $36.98 \pm 5.49$ | $36.54 \pm 5.68$ |
| True | False | tversky (fbank=3) | 5 | $50.81 \pm 11.99$ | $49.92 \pm 12.54$ |
| True | False | tversky (fbank=5) | 5 | $94.09 \pm 5.41$ | $93.91 \pm 5.22$ |
| True | False | tversky (fbank=8) | 5 | $100.00 \pm 0.00$ | $99.55 \pm 0.06$ |
| True | False | tversky (fbank=13) | 5 | $100.00 \pm 0.00$ | $99.59 \pm 0.11$ |
| True | False | tversky (fbank=21) | 5 | $100.00 \pm 0.00$ | $99.58 \pm 0.08$ |
| True | False | tversky (fbank=34) | 5 | $100.00 \pm 0.00$ | $99.58 \pm 0.07$ |
| True | False | tversky (fbank=55) | 5 | $100.00 \pm 0.00$ | $99.57 \pm 0.04$ |
| True | True | baseline | 5 | $87.35 \pm 0.11$ | $86.05 \pm 0.54$ |
| True | True | tversky (fbank=2) | 5 | $36.14 \pm 1.69$ | $36.09 \pm 2.19$ |
| True | True | tversky (fbank=3) | 5 | $51.82 \pm 6.07$ | $51.92 \pm 6.48$ |
| True | True | tversky (fbank=5) | 5 | $69.91 \pm 4.43$ | $69.68 \pm 4.72$ |
| True | True | tversky (fbank=8) | 5 | $82.00 \pm 2.80$ | $81.04 \pm 2.72$ |
| True | True | tversky (fbank=13) | 5 | $87.86 \pm 0.17$ | $86.17 \pm 0.48$ |
| True | True | tversky (fbank=21) | 5 | $89.42 \pm 0.07$ | $85.96 \pm 0.53$ |
| True | True | tversky (fbank=34) | 5 | $92.31 \pm 0.17$ | $86.80 \pm 0.37$ |
| **True** | **True** | **tversky (fbank=55)** | **5** | $94.66 \pm 0.07$ | $86.81 \pm 0.41$ |
| **False** | **False** | **baseline** | **5** | $100.00 \pm 0.00$ | $99.28 \pm 0.09$ |
| False | False | tversky (fbank=2) | 5 | $10.41 \pm 0.71$ | $10.44 \pm 1.13$ |
| False | False | tversky (fbank=3) | 5 | $10.15 \pm 0.57$ | $10.10 \pm 1.00$ |
| False | False | tversky (fbank=5) | 5 | $12.69 \pm 4.81$ | $12.47 \pm 4.36$ |
| False | False | tversky (fbank=8) | 5 | $30.80 \pm 9.63$ | $30.43 \pm 9.64$ |
| False | False | tversky (fbank=13) | 5 | $34.91 \pm 18.04$ | $34.71 \pm 17.82$ |
| False | False | tversky (fbank=21) | 5 | $52.81 \pm 16.38$ | $52.36 \pm 16.47$ |
| False | False | tversky (fbank=34) | 5 | $72.70 \pm 20.23$ | $71.42 \pm 20.85$ |
| False | False | tversky (fbank=55) | 5 | $96.24 \pm 8.41$ | $95.34 \pm 8.70$ |

## E.2 MODEL SELECTION

Table 22: Summary: Best Baseline and Tversky Models for Each Configuration

| Pretrained | Freeze | Best Baseline | | | Best Tversky | | | |
|---|---|---|---|---|---|---|---|---|
| | | Valid Acc (%) | Batch | Clip | Valid Acc (%) | Batch | Clip | FBank |
| True | False | $99.61 \pm 0.05$ | 256 | 10 | $99.62 \pm 0.06$ | 256 | 1 | 34 |
| True | True | $86.05 \pm 0.54$ | 256 | 10 | $86.85 \pm 0.20$ | 256 | 1 | 55 |
| False | False | $99.52 \pm 0.07$ | 32 | 1 | $99.54 \pm 0.09$ | 32 | 1 | 55 |

## E.3 EVALUATION OF BEST MODELS

Table 23: Summary: Best Baseline and Tversky Models for Each Configuration

| Pretrained | Freeze | Best Baseline | | | | | | Best Tversky | | | | | | |
|---|---|---|---|---|---|---|---|---|---|---|---|---|---|---|
| | | Train Acc (%) | Valid Acc (%) | Test Acc (%) | Batch | Clip | N | Train Acc (%) | Valid Acc (%) | Test Acc (%) | Batch | Clip | FBank | N |
| True | False | $100.00 \pm 0.00$ | $99.61 \pm 0.05$ | $99.56 \pm 0.04$ | 256 | 10 | 5 | $100.00 \pm 0.00$ | $99.62 \pm 0.06$ | $99.60 \pm 0.05$ | 256 | 1 | 34 | 5 |
| True | True | $90.24 \pm 0.27$ | $86.08 \pm 0.49$ | $86.64 \pm 0.14$ | 256 | 10 | 5 | $98.18 \pm 0.26$ | $86.80 \pm 0.23$ | $86.66 \pm 0.17$ | 256 | 1 | 55 | 5 |
| False | False | $100.00 \pm 0.00$ | $99.52 \pm 0.07$ | $99.54 \pm 0.04$ | 32 | 1 | 5 | $100.00 \pm 0.00$ | $99.54 \pm 0.09$ | $99.56 \pm 0.06$ | 32 | 1 | 55 | 5 |

# F    EXPERIMENT 103: [TVERSKY]RESNET-50 ON NABIRDS

ResNet-50 and TverskyResNet-50 were trained for 600 epochs with SGD (learning rate 0.03, momentum 0.9, weight decay $10^{-4}$, gradient clipping threshold 10) and a cosine annealing learning rate schedule. We used a batch size of 256 and dropout of 0.3. Input images were resized to $600 \times 600$ pixels and randomly cropped to $448 \times 448$ during training; at test time, a center crop of $448 \times 448$ was used. Training images were augmented with random horizontal flips, and all images were normalized using ImageNet mean and standard deviation. The Tversky projection layer used `product` and `ignorematch` as the intersection and difference reduction hyperparameters respectively, with feature bank sizes drawn from the Fibonacci sequence: 610, 987, 1597, 2584, 4181, 6765, and 10946. Each configuration was trained with multiple seeds, corresponding to different initializations of model parameters, shuffling of training data, and partitioning of train/validation data. 10% of the training split was used as a validation set. Test results are reported on the held-out standard NABirds test set. All accuracies are reported as mean $\pm$ std.

## F.1    HYPER-PARAMETER SEARCH

Table 24: Results for Batch Size=256, Gradient Clipping=10

| Pretrained | Freeze | Model | Seeds | Train Acc (%) | Valid Acc (%) |
|---|---|---|---|---|---|
| True | False | baseline | 3 | $99.97 \pm 0.01$ | $86.71 \pm 1.13$ |
| True | False | tversky (fbank=610) | 2 | $99.96 \pm 0.01$ | $85.83 \pm 1.19$ |
| True | False | tversky (fbank=987) | 2 | $99.97 \pm 0.01$ | $86.71 \pm 0.49$ |
| True | False | tversky (fbank=1597) | 3 | $99.97 \pm 0.02$ | $86.18 \pm 0.72$ |
| True | False | tversky (fbank=2584) | 3 | $99.98 \pm 0.01$ | $86.12 \pm 0.51$ |
| True | False | tversky (fbank=4181) | 3 | $99.98 \pm 0.01$ | $86.50 \pm 0.43$ |
| **True** | **False** | **tversky (fbank=6765)** | **2** | $99.97 \pm 0.02$ | $87.03 \pm 1.35$ |
| True | False | tversky (fbank=10946) | 3 | $99.96 \pm 0.02$ | $86.22 \pm 0.64$ |
| **True** | **True** | **baseline** | **3** | $79.33 \pm 0.25$ | $51.88 \pm 0.44$ |
| True | True | tversky (fbank=610) | 3 | $96.40 \pm 0.42$ | $48.21 \pm 1.27$ |
| True | True | tversky (fbank=987) | 3 | $98.67 \pm 0.34$ | $48.23 \pm 0.50$ |
| True | True | tversky (fbank=1597) | 2 | $99.20 \pm 0.29$ | $50.24 \pm 0.99$ |
| True | True | tversky (fbank=2584) | 2 | $99.30 \pm 0.06$ | $51.52 \pm 0.02$ |
| True | True | tversky (fbank=4181) | 3 | $99.49 \pm 0.08$ | $50.93 \pm 2.09$ |
| True | True | tversky (fbank=6765) | 3 | $99.56 \pm 0.08$ | $51.37 \pm 1.33$ |
| True | True | tversky (fbank=10946) | 3 | $99.49 \pm 0.10$ | $50.74 \pm 0.77$ |
| False | False | baseline | 3 | $99.99 \pm 0.01$ | $68.67 \pm 1.19$ |
| False | False | tversky (fbank=610) | 3 | $99.89 \pm 0.05$ | $70.41 \pm 0.73$ |
| False | False | tversky (fbank=987) | 3 | $99.92 \pm 0.05$ | $70.55 \pm 0.74$ |
| **False** | **False** | **tversky (fbank=1597)** | **3** | $99.93 \pm 0.04$ | $71.14 \pm 0.86$ |
| False | False | tversky (fbank=2584) | 3 | $99.93 \pm 0.00$ | $71.12 \pm 0.79$ |
| False | False | tversky (fbank=4181) | 3 | $99.89 \pm 0.02$ | $70.53 \pm 1.20$ |
| False | False | tversky (fbank=6765) | 2 | $99.91 \pm 0.02$ | $71.09 \pm 1.57$ |
| False | False | tversky (fbank=10946) | 3 | $99.91 \pm 0.06$ | $69.07 \pm 1.12$ |

## F.2 MODEL SELECTION

Table 25: Summary: Best Baseline and Tversky Models for Each Configuration

| Pretrained | Freeze | Best Baseline | | | Best Tversky | | | |
|---|---|---|---|---|---|---|---|---|
| | | Valid Acc (%) | Batch | Clip | Valid Acc (%) | Batch | Clip | FBank |
| True | False | $86.71 \pm 1.13$ | 256 | 10 | $87.03 \pm 1.35$ | 256 | 10 | 6765 |
| True | True | $51.88 \pm 0.44$ | 256 | 10 | $51.52 \pm 0.02$ | 256 | 10 | 2584 |
| False | False | $68.67 \pm 1.19$ | 256 | 10 | $71.14 \pm 0.86$ | 256 | 10 | 1597 |

## F.3 EVALUATION OF BEST MODELS

Table 26: Summary: Best Baseline and Tversky Models for Each Configuration

| Pretrained | Freeze | Best Baseline | | | | | | Best Tversky | | | | | | |
|---|---|---|---|---|---|---|---|---|---|---|---|---|---|---|
| | | Train Acc (%) | Valid Acc (%) | Test Acc (%) | Batch | Clip | N | Train Acc (%) | Valid Acc (%) | Test Acc (%) | Batch | Clip | FBank | N |
| True | False | $99.99 \pm 0.01$ | $86.71 \pm 1.13$ | $82.37 \pm 0.25$ | 256 | 10 | 3 | $100.00 \pm 0.01$ | $87.03 \pm 1.35$ | $82.96 \pm 0.07$ | 256 | 10 | 6765 | 2 |
| True | True | $90.92 \pm 0.24$ | $51.88 \pm 0.44$ | $40.25 \pm 0.28$ | 256 | 10 | 3 | $99.94 \pm 0.02$ | $51.47 \pm 0.10$ | $38.73 \pm 0.20$ | 256 | 10 | 2584 | 2 |
| False | False | $99.99 \pm 0.01$ | $68.67 \pm 1.19$ | $62.84 \pm 0.45$ | 256 | 10 | 3 | $99.97 \pm 0.01$ | $71.14 \pm 0.86$ | $65.20 \pm 0.26$ | 256 | 10 | 1597 | 3 |

## F.4 EXAMPLE OF LEARNED $\alpha$, $\beta$ AND $\theta$

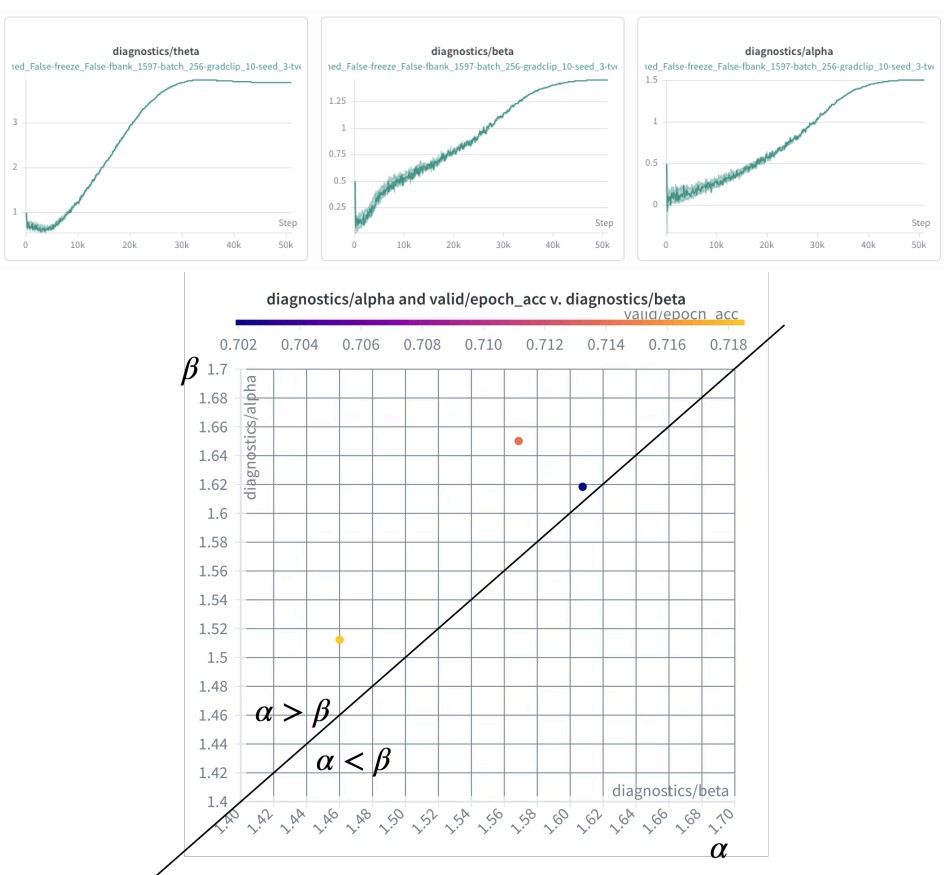

Figure 17: Top: Example of learned $\alpha$, $\beta$ and $\theta$ values over training iterations of a TverskyResNet-50 on NABirds trained from scratch. These parameters are learned via gradient descent alongside all other model parameters. Bottom: Comparison of final learned $\alpha$ (x-axis) and $\beta$ (y-axis) across 3 runs of the same configuration with different seeds; point color represents validation accuracy. In all three models $\alpha > \beta$, consistent with Tversky's theory that the distinctive features of instances are weighted more heavily than those of prototypes. While we observed $\alpha > \beta$ in several successful models, not all accurate models had this property.

## F.5 COMPUTATION RESOURCES

Table 27: Duration of model training sessions in wall-clock time (h:mm:ss), grouped by model type, GPU type, GPU count, and feature bank size (Tversky only). Note that the measured durations depend on various factors external to the architecture and hyperparameters of models, including I/O access time to network drives, which fluctuates with cluster usage. Std is reported as − when $N = 1$.

| Model | GPU | #GPU | fbank | $N$ | Mean | Std | Min | Max |
|---|---|---|---|---|---|---|---|---|
| Baseline | NVIDIA A100-SXM4-40GB | 4 | – | 3 | 28:32:18 | 15:55:15 | 10:09:22 | 37:56:42 |
| Baseline | NVIDIA A100-SXM4-80GB | 4 | – | 5 | 10:00:14 | 0:29:02 | 9:15:07 | 10:28:44 |
| Baseline | NVIDIA RTX A6000 | 4 | – | 1 | 10:45:45 | – | 10:45:45 | 10:45:45 |
| Tversky | NVIDIA A100-SXM4-40GB | 3 | 4181 | 1 | 9:14:14 | – | 9:14:14 | 9:14:14 |
| Tversky | NVIDIA A100-SXM4-40GB | 3 | 6765 | 1 | 9:24:51 | – | 9:24:51 | 9:24:51 |
| Tversky | NVIDIA A100-SXM4-40GB | 4 | 2584 | 1 | 9:57:01 | – | 9:57:01 | 9:57:01 |
| Tversky | NVIDIA A100-SXM4-40GB | 4 | 4181 | 1 | 10:00:42 | – | 10:00:42 | 10:00:42 |
| Tversky | NVIDIA A100-SXM4-40GB | 4 | 6765 | 1 | 10:11:54 | – | 10:11:54 | 10:11:54 |
| Tversky | NVIDIA A100-SXM4-40GB | 4 | 10946 | 1 | 10:01:52 | – | 10:01:52 | 10:01:52 |
| Tversky | NVIDIA A100-SXM4-80GB | 2 | 610 | 2 | 9:12:21 | 0:27:41 | 8:52:46 | 9:31:56 |
| Tversky | NVIDIA A100-SXM4-80GB | 2 | 987 | 1 | 9:46:35 | – | 9:46:35 | 9:46:35 |
| Tversky | NVIDIA A100-SXM4-80GB | 2 | 1597 | 1 | 8:34:17 | – | 8:34:17 | 8:34:17 |
| Tversky | NVIDIA A100-SXM4-80GB | 2 | 2584 | 1 | 9:31:55 | – | 9:31:55 | 9:31:55 |
| Tversky | NVIDIA A100-SXM4-80GB | 2 | 10946 | 1 | 8:24:21 | – | 8:24:21 | 8:24:21 |
| Tversky | NVIDIA A100-SXM4-80GB | 4 | 610 | 7 | 9:34:20 | 0:24:56 | 9:12:26 | 10:08:32 |
| Tversky | NVIDIA A100-SXM4-80GB | 4 | 987 | 4 | 9:41:57 | 0:35:41 | 9:12:16 | 10:25:39 |
| Tversky | NVIDIA A100-SXM4-80GB | 4 | 1597 | 5 | 9:21:46 | 0:28:12 | 8:40:10 | 9:52:03 |
| Tversky | NVIDIA A100-SXM4-80GB | 4 | 2584 | 6 | 9:46:25 | 0:29:38 | 9:11:34 | 10:20:07 |
| Tversky | NVIDIA A100-SXM4-80GB | 4 | 4181 | 6 | 9:27:43 | 0:27:41 | 8:45:03 | 10:09:11 |
| Tversky | NVIDIA A100-SXM4-80GB | 4 | 6765 | 4 | 9:28:17 | 0:33:22 | 8:56:45 | 10:08:13 |
| Tversky | NVIDIA A100-SXM4-80GB | 4 | 10946 | 7 | 9:26:21 | 0:43:17 | 8:38:53 | 10:28:57 |
| Tversky | NVIDIA H200 | 2 | 987 | 1 | 6:25:57 | – | 6:25:57 | 6:25:57 |

