# OpenReview forum: "Tversky Neural Networks: Psychologically Plausible Deep Learning with Differentiable Tversky Similarity"
_ICLR.cc/2026/Conference — ICLR 2026 Poster_

### Official Review · Reviewer_c2vD · 2025-10-29

**Soundness:** 3
**Presentation:** 4
**Contribution:** 4
**Rating:** 8
**Confidence:** 4

**Summary:**

The authors propose how to use Tversky similarity (from cognitive psychology) instead of the geometric model of similarity, which is usually used in deep learning. They perform experiments on text and image domains.

**Strengths:**

* I really like the idea of using Tversky similarity to create layers for neural networks. Tversky’s theory is very logical and simple and I think that it has a huge potential for the neural network domain.
* The paper is well-written and easy to follow. It is beneficial that the authors presented a small XOR example at the beginning.
* The salience experiment was interesting.

**Weaknesses:**

* The paper lacks a clear related work section. While some works are described in the introduction, it would be good to include a dedicated section. Also, there are works that use Tversky’s similarity model or methods inspired by it for different purposes in the deep learning domain, and it would be beneficial to mention these works and acknowledge that using Tversky’s model of similarity is not completely new in deep learning - even if for different purposes.
* “This lack of interpretability, even in this simple domain without background textures, represents a significant limitation of prior approaches.” - the authors say that their approach improves interpretability. While it is visible that for MNIST (simple objects, with large differences) the prototypes look reasonable, I am not so sure that they would be that pronounced for the second dataset used in the study - regarding different breeds of birds.
* For the vision experiments, it would be nice to include the results for a ViT as well - to represent two leading architecture families in vision.

**Questions:**

* “Some initializations of prototypes and features lead to convergence failure” - are normal and orthogonal initialization the ones that the authors refer to?
* Why didn’t the authors present the example graphical results for NABirds (like the ones in Fig. 2 for MNIST)?
* Also, see the weaknesses.

---

> ### Author Response · Authors · 2025-12-03
> **Response to Reviewer c2vD**
>
> ## Related work
> - We preferred to discuss the relevant prior work in the introduction section because of the novelty of our approach.
> - We will expand our discussion to include [1], which is the only prior work we could find that uses Tversky Similarity in machine learning, but for a purpose of estimating image segmentation quality, wherein pixels are seen as the set of features of the ground truth and the predicted segmentation mask.
>
> ## Prototype Visualization for more complex domains
> - We defer this to future work.
>
> ## Experiments with more neural architectures such as VIT
> - We defer this to future work
>
> ## Convergence failure due to initialization
> - We are referring to specific initialization conditions where some features are not in the set of any prototype or data points, which could emerge from any initialization method, although we found uniform initialization to work more frequently for XOR. See Figure 6 and Table 5. Also see our response to Reviewer wB3R.
> In practice with larger datasets (MNIST and NABirds), we did not encounter this issue, and found the training to be stable.
>
> ## Prototype Visualization for NABirds
> As we discussed in section 2.5, note that our prototype visualization method results in higher parameter count and higher computation requirements for large image domains, and large class counts (e.g. 555 448x448 visual prototypes for NABirds). We are currently running experiments applying this technique to NABirds. We will update the final version of the manuscript with the results.
>
>
>
> ## Additional References
> [1] Rahnama, J., & Hüllermeier, E. (2020, June). Learning tversky similarity. In International Conference on
> Information Processing and Management of Uncertainty in Knowledge-Based Systems (pp.
> 269-280). Cham: Springer International Publishing.

---

### Official Review · Reviewer_ymVH · 2025-10-30

**Soundness:** 3
**Presentation:** 2
**Contribution:** 3
**Rating:** 6
**Confidence:** 4

**Summary:**

The paper proposes an alternative to the commonly geometric formulation of similarity through using Tversky similarity. The authors propose a differentiable formulation of representations for this purpose, along with an implementation thereof (Tversky Projection neural layer). In addition, an approach for interpreting Tversky networks through visualization of projection layers in the data domain are proposed. Experiments on the XOR, Penn Treebank text data, and MNIST are performed to demonstrate the method.

**Strengths:**

- Clear and well-founded scientific motivation for the proposed novel methods. The idea of using Tversky similarity for deep learning is to my knowledge a novel contribution.
- Empirical confirmation of the proposed neuralized Tversky approach, in both synthetic experiments and large-scale studies
- The paper includes an honest and balanced discussion section and transparently raises limitations throughout the paper.
- The experiments appear reproducible and are for the most part clearly described (see below).

**Weaknesses:**

- While I like the motivation of Section 3.1 (XOR), the section and associated Figure 1 read rushed and can be confusing in its current state. I think the main take-home is that the Tversky projection layer can in principle learn the XOR function. I would suggest focusing on introducing the problem clearly (which is currently done in the caption), and then discuss some of the limitations encountered during optimization, i.e., convergence and hyperparameter selection.
- While I think the experiments are overall good, they remained superficial at times. Some quite interesting points were raised in the discussion, e.g. the idea of editing prototypes for increased robustness, which would have been great to demonstrate also empirically.
- Some claims regarding interpretability are not clearly supported by evidence, e.g. “handwriting, such as lines and curves, more clearly than those learned by linear projection layers” (l. 413) and “Tversky Projection layers’s parameters are far more interpretable than the ones of the contemporary fully connected layer“ (l. 217). A more fair comparison could be to compare attributions in input space, e.g. [Mon18] as the Tversky Projection layers explicitly tie the projection to being decodable from the input.
- The description of the proposed visualization technique in Section 2.5 should be more clear, e.g. through the use of formulas or a conceptual visualization, e.g. moving Fig. 5 to the main alongside a more clear description of how the method works. For example, the sentence “limitation that parameters specified in data-space are typically larger in size than their original counterparts, which increases the effective number of trainable parameters.“ (l. 211-213) is difficult to parse, i.e. how are the projection parameters specified, is the visualization technique trained or is this a post-hoc interpretability approach?


**Comment**
- I believe the proposed framework could be quite useful in the context of textual similarity, where it was shown that standard encoders and cosine similarity do not match human similarity judgments very well [Rei19]. Follow-up work on interpreting these dot-product embeddings could reveal quite simplistic feature matching strategies, even after aligning predictions to human similarity judgements [Vas24]. Here a Tversky layers may provide a more effective similarity readout.


**References**
- [Rei19] Reimers, N., & Gurevych, I. (2019, November). Sentence-BERT: Sentence Embeddings using Siamese BERT-Networks. In Proceedings of the 2019 Conference on Empirical Methods in Natural Language Processing and the 9th International Joint Conference on Natural Language Processing (EMNLP-IJCNLP) (pp. 3982-3992).
- [Vas24] Vasileiou, A., & Eberle, O. (2024, June). Explaining Text Similarity in Transformer Models. In Proceedings of the 2024 Conference of the North American Chapter of the Association for Computational Linguistics: Human Language Technologies (Volume 1: Long Papers) (pp. 7852-7866).
- [Mon18] Montavon, G., Samek, W., & Müller, K. R. (2018). Methods for interpreting and understanding deep neural networks. Digital signal processing, 73, 1-15.

**Questions:**

- Assuming contextualized models, would they be able to learn Tversky-type similarity?
- The notion of feature bank Ω could be communicated more clearly in Section 2.3. From Section 2.2 these are feature vectors $f_k$ and Ω parametrizes the function $f$, so $f$ essentially an embedding layer (as also mentioned more clearly later in Sec. 2.4.1).
- “For our vision experiments, no tuning was performed” (l. 459): No tuning *of hyperparameters* if I read Section 3.3 correctly, is that correct?

**Editing*
- “$α, β, andθ$” (l. 181)
- “fullly” (l. 217)
- “weights are initialized from ImageNet” (l. 327) - ImageNet -> ResNet?

**Details Of Ethics Concerns:**

None.

---

> ### Author Response · Authors · 2025-12-03
> **Response to reviewer ymVH**
>
> We thank the reviewer for their thoughtful comments.
>
>
> ## Section 3.1 (XOR)
> We want to clarify that the purpose of this section is to show that Tversky Projection can model XOR by construction, and through gradient descent (with only 3 non-zero data points), and to show its sensitivity to hyperparameters and its failure modes in this simple domain. We expect that readers will find Figure 1 very helpful in understanding how Tversky Neural Networks work. This section also references Appendix C which reports convergence rates for various hyperparameter combinations, and illustrate failure modes (Figure 6).
>
> ## Prototype Editing
> The explainability of Tversky prototypes led us to formulate the hypothesis that our visualization method could be combined with manual interpretable editing. We left this hypothesis in the discussion section to motivate future work.
>
> ## Explainability and future visualization
> We want to clarify the reviewer's note that "Tversky Projection layers explicitly tie the projection to being decodable from the input." may have resulted from a misunderstanding of section 2.5. We have updated Figure 5 to enhance clarity.
> Our prototype visualization method is NOT tied to Tversky Neural Networks; it equally applies to classical linear projections.
>
> ## Formalization of our proposed prototype visualization technique
> We thank the reviewer for this great idea. We have updated Figure 5, and introduced some formalism to enhance clarity.
>
>
> ## NOT Post-hoc interoperability
> The visualization technique applies at training time. It allows learning the projection parameters in the input domain. This method is strictly applied at training time, resulting in projection parameters that can be visualized in the same domain as the input data.
>
> ## Other clarifications
> - "Assuming contextualized models, would they be able to learn Tversky-type similarity?".
> We kindly request that the reviewer clarifies this question. Assuming that the reviewer means contextualized word embeddings, as produced by models such as BERT, which produce an encoding of tokens that is "contextualized" in the sense that the embedding also includes information extracted from other tokens in the same context, yes, Tversky neural modules can be used in such context. For instance, our TverskyGPT2 models produce contextualized token embeddings that are classified into discrete tokens using a Tversky projection comparing them to prototypes consisting of all tokens in the token dictionary.
>
> - Feature bank $\Omega$
> The feature bank is a set of $|\Omega|$ vectors, which we implement as a torch Embedding layer.
>
> - "For our vision experiments, no tuning was performed (l. 459): No tuning of hyperparameters
> if I read Section 3.3 correctly, is that correct?"
> Yes, that is correct. This is was contrast to our experiments with GPT2 and TverskyGPT2 where multiple hyperparameters were tried. Note that GPT2 and ResNet-50 have benefited from years of hyperparameter tuning for various tasks. We encourage future work to perform various tuning experiments on Tversky neural networks to develop our understanding of when they work best.
>
> - "Editing /α, βandtheta (l. 181)
>     - Fixed
>
> - Editing / fullly (l. 217)
>     - Fixed
>
> - ImageNet (l. 327) - ImageNet -> ResNet?-50 trained on ImageNet"
>     - Fixed

---

### Official Review · Reviewer_XcN8 · 2025-10-31

**Soundness:** 4
**Presentation:** 4
**Contribution:** 4
**Rating:** 8
**Confidence:** 4

**Summary:**

The paper introduces a new perspective on classification via prototype learning, where classifier decisions are made by taking the argmax over similarities to class prototypes.  Similarity is defined as an asymmetric match function between the object and a prototype over a learned feature bank whose dimensionality can be set.  While asymmetric similarity has been used in some prior applications (e.g., nearest neighbor classification based on engineered features), to my knowledge it has not been used in end-2-end CNN training, and the application is novel.

**Strengths:**

Strengths:
* Strong originality of entire prototype learning framework based on learned asymmetric comparison, which can be integrated in end-to-end training of various deep architectures.
* The asymmetry weights, prototype dimensionality, and the feature bank are treated as learnable (free) parameters.
* Demonstrates improved performance on several benchmarks, including NABirds, indicating applicability to problems with hundreds of classes. While there is no demonstration of extension to thousands of classes, in this case I do not see as a limitation, because one of the central future applications of this training approach may be application to classification in relatively small domains with highly similar categories.
* A strength of the approach, whichcould be better discussed, is the potential to constrain the directionality of the asymmetry weights (e.g., setting β>α when omissions matter more, α>β when extras hurt more) depending on domain. For domains with low intra-class variance where 'core' features count (quasi necessary), penalizing omissions  of object vs. prototype might be more sensible; the converse for broader, more fuzzy categories.

**Weaknesses:**

(Minor) Weaknesses:

* It would be nice to have a small ablation experiment isolating the specific contribution of asymmetry by setting the asymmetry weights to zero and learning only the overlap (intersection) term in Equation 1.
* Can be extended to allow learning only the asymmetry terms with the overlap term to zero. Taken together we can understand if performance gains rely mainly on similarities or differences.

**Questions:**

It would be good to know if the Tversky head is effective as a classifier when attached to a backbone not trained for classification (e.g., a self-supervised backbone) and if in that case it would also outperform a baseline linear readout.

---

> ### Author Response · Authors · 2025-12-03
> **Response to Reviewer XcN8**
>
> We thank the reviewer for their thoughtful comments.
>
> ## Ablation on the specific contribution of asymmetry
> We want to recall that our primary motivation is the incorporation of (Tversky, 1977)' psychological model of similarity into deep learning, in which asymmetry is a desired an necessary component. This is the basis of our grounding in psychology. Nonetheless, we find the ablation proposed by the reviewer interesting, and will defer it to future work.
>
>
>
> ## Backbones not trained for classification
> This is another excellent research question to explore in future work. For instance, an experiment could compare the accuracy of a baseline sentiment classification head with its Tversky counterpart when attached to a pre-trained Bert Model.

---

### Official Review · Reviewer_wB3R · 2025-10-31

**Soundness:** 3
**Presentation:** 3
**Contribution:** 3
**Rating:** 6
**Confidence:** 3

**Summary:**

This paper challenges the geometric model of similarity (dot-product/ cosine similarity) largely used in deep learning, proposing instead Tversky's psychologically-grounded similarity measure that permits asymmetry and other violations of metric axioms observed in human similarity judgment. The key contribution is a differentiable version of Tversky's similarity and corresponding neural layers that can replace dense layers in standard architectures. The approach is based on the introduced dual representation of an object: objects are represented both as vectors and as sets of features from a learnable finite universe, where an object's set comprises those feature vectors with which it has positive dot product. This set-theoretic representation enables measuring similarity through feature set intersections and differences, forming the basis for Tversky Similarity and Projection layers. The authors demonstrate that: (1) a single Tversky layer can model XOR, impossible for linear layers; (2) Tversky layers can improve performance in GPT-2 language modeling (PTB dataset) and ResNet-50 image classification (MNIST, NABirds). Qualitative analyses reveal interpretable properties: salience scores (sum of dot-products of an object's vector with all features in its set) align with "goodness of form," learned (input-level) prototypes are human-interpretable, and semantic fields (e.g., adjectives, verb forms) emerge from set algebra. Additionally, they introduce a data-domain visualization method for projection parameters, though this approach scales poorly.

The work is conceptually novel, well-written, and really promising but could benefit from modern architecture evaluation, clearer differentiability explanation, and stronger scalability analysis.

**Strengths:**

- Original and well-motivated idea connecting human similarity judgments to machine-learned representations. Clear novelty of First differentiable implementation of Tversky similarity for neural networks.
- The paper is well-written, clearly structured, and engaging to read. I learned a lot.
- The XOR demonstration (Section 3.1, Figure 1) with its thorough validation convincingly shows that a single Tversky projection can model non-linear decision boundaries without composition with activation functions. This is a genuine advantage over linear layers.
- Demonstrates potential in both vision and language modeling contexts.
- Qualitative analysis really interesting.
- Acknowledges computational cost of data-domain visualization (Figure 5) and it is honest about the limitations.

**Weaknesses:**

- Gradient flow through indicator functions: Equations 2-5 rely on indicator functions [a·fₖ > 0] which have zero derivative almost everywhere. It's unclear how gradients propagate during training—are you using straight-through estimators, sigmoid approximations, or another approach? Providing implementation details or code would help to understand you work.
- Convergence of training: Tables 4-7 show convergence failure rates of 47-77% across many hyperparameter settings, with some producing NaN results. Understanding what causes these failures would strengthen the work. How did convergence look for the other main experiments?
- Evaluation on larger, more recent benchmarks would strengthen claims. Current experiments use MNIST (toy dataset), PTB (1M tokens from 1993), and NABirds (48K images with domain overlap with ImageNet birds), which are ideal for a proof of concept. However, testing on standard modern benchmarks like ImageNet, CIFAR-100, or WikiText-103 would better demonstrate competitiveness against well-tuned baselines.
- Similarly, validation on current architectures would show broader applicability. While ResNet-50 and GPT-2 small are solid choices for controlled experiments, evaluating on Vision Transformers (ViT, CLIP, DINOv2) or larger language models would support the "state-of-the-art" framing (L82).
- Multiple seeds are reported for XOR experiments, but it's unclear if main experiments used multiple runs. Would it be possible to comment on this issue or if available report the means and standard deviations?
- Tied vs untied feature banks remain unexplored. While prototype tying is ablated in Table 1, feature bank sharing is always enabled. Testing both configurations would be informative.
- Data-domain visualization might be impractical at scale. For NABirds with 555 classes, input-space parameterization would require 83.5M parameters (555×224×224×3) compared to 1.1M for standard approaches—a 75× increase. Could the authors comment on the scalability constraints and potential overfitting risks?
- Reporting accuracies: Using relative improvements (e.g., 24.7%) rather than absolute accuracy points can sometimes overemphasize small differences. Both would be informative.
- Methods like ProtoPNet and ProtoTree (prototype learning domain) seem related to the data-domain visualization contribution but aren't discussed in the introduction or chapter 2.5. Could the authors comment on that?

**Questions:**

- Please see questions already stated in weaknesses :)
- Methods:
    - As the author’s mention in the discussion, the size of  $\Omega$  is hyperparameter and needs to be tuned as everything else. However, do the authors have an intuition if there could be a relationship between the number of prototypes and the universe size?
    - Question about the set representations: A feature $f_k$ of the learnable finite universe is in the set of object $x$ if their dot-product is larger than zero. What does it then mean to include into this set representation a feature $f_j$  with $x\cdot f_j = 1e^-10$ and a feature $f_q$ with $x\cdot f_q = 1e^-10$?
- Experiments
    - Tversky layer can model XOR: In Tables 4-7 we can an observe rather small convergence ratios, could the authors comment on that?
    - Language-modeling: In Table 1, one can observe that only replacing the prediction head with a tversky head results in more features and parameters compared to replacing all projection layers with Tversky projections. This makes sense as the projection head itself must learn to represent all the semantic structure needed for predicting the next token. When all projection layers are replaced, the encoder and attention blocks co-adapt with the new similarity function, so the features can specialize efficiently, and fewer prototypes are sufficient. However, could you elaborate on the feature number and parameter number if we compare tversky-all with and without prototyp tying? Why do we observe such a difference?
    - Image classification:
        - Would it be possible to show some image classification results with some a pretrained Vision Transformer. E.g., a vision-encoder of a CLIP model, or Dinov3 model (available on huggingface)?
        - Why are the pretrained and frozen setting so bad in performance?

---

> ### Author Response · Authors · 2025-12-03
> **Response to reviewer wB3R Part 1**
>
> Thank you for your thoughtful review and constructive feedback. We address your main concerns below:
>
>
> ## Differentiability
> Feature measures in objects and prototypes (dot product) are only defined when such dot-product is positive. In that positive domain, a feature measure is continuous and differentiable. Features may dynamically enter or leave an object's or prototype's set during training as the neural network parameters update.
>
> ## Gradient Flow and Straight-Through Estimators
> The indicator functions in Equations 2-5 make Tversky layers sparsely activated neural modules comparable to the gating network of (MoEs) layers [1, 2, 3], with the indicator function behaving like a hard router. TNNs sparse activations makes them theoretically subject to the following potential issues, which have also been explored in MoEs.
> - (a) Some features could be disproportionately more active than others. (comparable to the "unbalanced expert load" issue leading to "routing collapse" in MoEs [4].
> - (b) Some features or prototypes could remain permanently inactive and always have 0 gradient. This is similar to the "dead expert" issue in MoEs [5].
>
> Recent work has explored straight-through estimators [5], load balancing losses [4], and top-k routing [6] to address the analogous issues in MoEs.
>
> Regarding (a), we do not believe that the feature use imbalance is an inherent modeling issue in Tversky Neural Networks because we do not expect the distribution of features in sets of stimuli to be uniform. Future work could explore potential implications of this imbalance in the context of sparsity-based hardware optimization mechanisms.
>
> Regarding (b), while monitoring prototype and feature gradient magnitudes over training iterations, we did not notice the "dead feature" or "dead prototype" issue in our computer vision and language modeling experiments. However, we noticed and documented this issue in the XOR domain, which only has 4 data points (See Figure 6, bottom row).
>
> We appreciate the reviewer's attention to this detail, and acknowledge that future work should assess the prominence of issue (b) in realistic large datasets, and investigate possible solutions if such problem exists, including straight-through estimators.
>
> We also recall that there are several documented sparsely activated neural networks including MoEs[1] and H-nets[7] that do not apply straight-through estimators and that are successful nonetheless.
>
> ## Convergence rates
> - The results in Table 4-7 include many settings designed to test failure modes of Tversky Projections, and its sensitivity to hyperparameter in the XOR domain. For instance, the convergence probability is very low when the feature bank size is only 1. These experiments also allowed us to uncover the "dead feature/prototype" issue which could occur in very small datasets (e.g. XOR with only 3 non-zero data points. See Figure 6, bottom row).  We did not encounter such issue in our larger-scale experiments.
> - Also see new results in appendix I. low-standard deviations (as the feature bank size increases) indicates consistent convergence.
>
> ## Dataset and Architecture Scope
> We believe that our experiments with our chosen datasets and baseline models are sufficient to support our hypotheses. However, we acknowledge that future work should explore larger datasets and other types of models. Expanding to all possible architectures and datasets would unnecessarily broaden the scope of this initial paper.
>
> ## Feature Bank Configuration for tversky-all
> - In the 'tversky-all' configuration, the feature bank is shared across all Tversky projection layers, including decoder output projection modules, and the language modeling head (See Line 288).
> - We have updated the tables in Appendix D to clarify this. (Note that in Appendix D, the tie-fbank column previously refered to tying feature banks to prototypes, which we did not end up exploring).
>
> ## Scalability of our data-domain prototype visualization
> We explained the limitations of this approach in section 2.5. Projection parameters specified in input-domain require more parameters than their vector-space specification, and are forwarded through the neural network along each data batch during training. We also clarify that this limitation doesn't apply at inference time as the data-domain prototypes can be forwarded once and reused for inference purposes.
>
> ## ProtoPNet and ProtoTree
> Both of these methods learn latent patch representations that are compared to convolutional feature maps to infer classification evidence. Our method is different in that we seek to visualize the parameters of projection layers, including fully connected layers and Tversky projection layers, in input space, which ProtoPNet and ProtoTree cannot do. Our method works for any projection layer at any depth. ProtoPNet and ProtoTree are restricted to convolutional feature map patches. We will contrast these prior works with our method in the final version of our manuscript.

---

> ### Author Response · Authors · 2025-12-03
> **Response to reviewer wB3R Part 2 & References**
>
> ## Multiple Seeds
> Added Appendix I with new experiment results for MNIST including multiple seeds, hyperparameter search and cross validation.
> The standard deviation of model accuracies (e.g. as a function of feature bank size) are also indicative of training stability. (note that these experiments use a different set of hyperparameters than the previously reported results, and overall improve the performance of both the baseline and Tversky models on MNIST). We will discuss this in the main text in the final version of the manuscript.
>
> ## Number of prototypes vs feature bank size
> We do not have a formal hypothesis on how these quantities may relate. We believe that their relation depends on the complexity of the dataset and its taxonomy.
>
> ## Set representation and feature measure
> In the example provided by the reviewer, both $f_j$ and $f_q$ would be included in the set of features of x, but with very low salience 1e-10. Our set representation is a "soft" set representation in that each membership corresponds to a positive measure value which varies by feature.
>
>
> ## Language Modeling Head parameter count
> A Tversky Projection layer always has more parameter than an equivalent single fully connected layer with equal input and equal output dimensions. This is because the fully connected layer parameters consist of prototype vectors whereas the Tversky projection also requires a feature bank. However Tversky Neural Networks can be more parameter efficient because in addition to any intrinsic modeling efficiency, (1) they can replace en entire MLP stack, and (2) multiple Tversky projection and similarity layers can share their feature banks.
> Please also note that in the tversky-all models, the feature bank is shared across all Tversky projection layers.
>
>
> ## Image Classification / frozen pretrained backbone
> Our hypothesis is that the lower performance in this setting is due to the domain distribution shift. Updating the pretrained weights results in better adaptation.
>
>
>
> ## Additional References
> - [1] Jacobs, R. A., Jordan, M. I., Nowlan, S. J., & Hinton, G. E. (1991). Adaptive mixtures of local experts. Neural computation, 3(1), 79-87.
> - [2] Shazeer, N., Mirhoseini, A., Maziarz, K., Davis, A., Le, Q., Hinton, G., & Dean, J. (2017). Outrageously large neural networks: The sparsely-gated mixture-of-experts layer. arXiv preprint arXiv:1701.06538.
> - [3] Fedus, W., Zoph, B., & Shazeer, N. (2022). Switch transformers: Scaling to trillion parameter models with simple and efficient sparsity. Journal of Machine Learning Research, 23(120), 1-39.
> - [4] Wang, L., Gao, H., Zhao, C., Sun, X., & Dai, D. (2024). Auxiliary-loss-free load balancing strategy for mixture-of-experts. arXiv preprint arXiv:2408.15664.
> - [5] Dai, D., Deng, C., Zhao, C., Xu, R. X., Gao, H., Chen, D., ... & Liang, W. (2024). Deepseekmoe: Towards ultimate expert specialization in mixture-of-experts language models. arXiv preprint arXiv:2401.06066.
> - [6] Lv, M., Su, Z., Pan, L., Xiong, Y., Lin, Z., Chen, H., ... & Hu, S. (2025, November). DSMoE: Matrix-Partitioned Experts with Dynamic Routing for Computation-Efficient Dense LLMs. In Proceedings of the 2025 Conference on Empirical Methods in Natural Language Processing (pp. 19722-19733).
> - [7] Hwang, S., Wang, B., & Gu, A. (2025). Dynamic chunking for end-to-end hierarchical sequence modeling. arXiv preprint arXiv:2507.07955.

---

### Author Response · Authors · 2025-12-03
**Summary of changes made to the PDF**

# Summary of changes made to the PDF
- Improved Figure 5 and introduced more formalism to explain our data-domain projection parameter visualization mechanism per our discussion with reviewer ymVH
- Fixed typos mentioned by ymVH
- Updated tables in Appendix D to remove the confusion about tversky-all noticed by reviewer wB3R
- Fixed error in Equation 5
- Added Appendix I with results for multiple seed runs across various hyperparameters for MNIST as requested by reviewer wB3R
- Note. We will make additional enhancements in the final version of our manuscript, including:
    - Additional clarifications based on reviewer's helpful feedback, which we discussed here.
    - Additional references we presented here
    - Results of our prototype visualization method for NABirds (visual representation of learned bird species prototypes). Experiments in progress.
    - Discussion of multiple seed run results in the main text.
    - Any constructive feedback from ACs

---

### Meta-Review · Area_Chair_rs7n · 2026-01-05

**Summary:**

The reviewers broadly agree that the paper presents a novel, well-motivated, and technically sound contribution, introducing an original and conceptually compelling differentiable formulation of Tversky similarity for deep learning. The main concerns focused on optimization behavior in small-scale settings (e.g., XOR), clarity around gradient flow through indicator functions, and the limited scope of empirical evaluation on modern datasets and architectures. The rebuttal satisfactorily addressed the key methodological and implementation-level questions, providing plausible explanations and additional analyses that alleviate the most critical doubts about technical correctness. While some concerns regarding scalability, broader benchmarking, and the largely qualitative nature of interpretability evidence remain, these are best viewed as limitations of experimental scope rather than fundamental flaws. Overall, the paper makes a meaningful and original contribution, and the remaining issues do not outweigh its strengths, leading to a cautiously but clearly positive recommendation.

**Reviewer Concerns:**

Addressed concerns:
The rebuttal effectively addressed most major technical concerns raised by the reviewers. In particular, questions regarding differentiability and gradient flow through indicator functions were clarified with concrete implementation details and analogies to sparsely activated modules, and convergence issues observed in small-scale XOR experiments were reasonably explained by data scarcity and hyperparameter sensitivity. Design choices around feature banks, prototype sharing, and parameter efficiency were also clarified, resolving several implementation-level ambiguities. Overall, these responses alleviated the primary doubts about technical correctness and soundness, and would likely lead some reviewers to maintain or slightly strengthen their original assessments.

Outstanding concerns:
A few aspects remain open for future improvement, most notably the limited scope of empirical evaluation on larger and more modern benchmarks and architectures, as well as the largely qualitative nature of the interpretability evidence. While additional analyses would strengthen claims about scalability and generality, these issues mainly reflect the current experimental scope rather than fundamental weaknesses of the proposed approach. They do not detract from the conceptual novelty, technical validity, or core contributions of the work, which overall remains slightly but clearly on the accept side.

**Reviewer Scores:**

Overall, Reviewer wB3R is likely to maintain their original score, or at most make a very slight upward adjustment, as the rebuttal effectively addressed the main technical questions on differentiability, gradient flow, convergence behavior, and parameterization choices, but did not fully resolve concerns regarding scalability, modern benchmarks, and the strength of empirical validation. Reviewer XcN8 would likely maintain a strong accept-level score, as the paper was already viewed as technically solid and well motivated, and the rebuttal further reinforced this positive assessment through clarifications and consistent positioning of the contribution. Reviewers ymVH and c2vD are likely to keep their original scores, since their remaining concerns—related to experimental depth, interpretability evidence, and broader architectural coverage—reflect limitations of scope and evaluation rather than fundamental weaknesses of the method. Taken together, the rebuttal improves overall confidence in the technical validity and framing of the work, leading to a slightly more favorable overall score distribution that supports an acceptance decision.

---

### Decision · Program_Chairs · 2026-01-26

Accept (Poster)